# dParallel:
# Learnable Parallel Decoding for dLLMs

**Zigeng Chen, Gongfan Fang, Xinyin Ma, Ruonan Yu, Xinchao Wang**[*]
National University of Singapore
zigeng99@u.nus.edu, xinchao@nus.edu.sg

## Abstract

Diffusion large language models (dLLMs) have recently drawn considerable attention within the research community as a promising alternative to autoregressive generation, offering parallel token prediction and lower inference latency. Yet, their parallel decoding potential remains largely underexplored, as existing open-source models still require nearly token-length decoding steps to ensure performance. To address this, we introduce dParallel, a simple and effective method that unlocks the inherent parallelism of dLLMs for fast sampling. We identify that the key bottleneck to parallel decoding arises from the sequential certainty convergence for masked tokens. Building on this insight, we introduce the core of our approach: certainty-forcing distillation, a novel training strategy that distills the model to follow its original sampling trajectories while enforcing it to achieve high certainty on masked tokens more rapidly and in parallel. Extensive experiments across various benchmarks demonstrate that our method can dramatically reduce the number of decoding steps while maintaining performance. When applied to the LLaDA-8B-Instruct model, dParallel reduces decoding steps from 256 to 30 on GSM8K, achieving an 8.5× speedup without performance degradation. On the MBPP benchmark, it cuts decoding steps from 256 to 24, resulting in a 10.5× speedup while maintaining accuracy. Our code is available at https://github.com/czg1225/dParallel

## 1 Introduction

Diffusion large language models (dLLMs) (Yu et al., 2025a; Zhang et al., 2025; Yi et al., 2024) have emerged as a promising alternative to autoregressive LLMs (Achiam et al., 2023; Bai et al., 2023). By leveraging bidirectional attention, they overcome the sequential generation bottleneck and enable parallel, random-order text generation, offering the potential for substantial improvements in inference efficiency. This potential has already been demonstrated in proprietary models such as Mercury (Labs et al., 2025), Gemini-Diffusion, and Seed-Diffusion (Song et al., 2025).

However, realizing this parallelism in existing open-source dLLMs remains challenging. Open implementations such as LLaDA (Nie et al., 2025; Zhu et al., 2025) and Dream (Ye et al., 2025), still require a number of decoding steps proportional to the sequence length to maintain generation quality, resulting in limited inference efficiency. Many recent efforts have attempted to accelerate dLLMs. Some approaches (Ma et al., 2025; Liu et al., 2025; Wu et al., 2025; Hu et al., 2025) reduce the time cost per decoding step by enabling KV caching. Other works (Israel et al., 2025; Wei et al., 2025; Li et al., 2025a;b; Gwak et al., 2025; Ben-Hamu et al., 2025) focus on optimizing parallel sampling algorithms to accelerate inference by reducing the necessary decoding steps. Despite these advancements, existing methods have yet to fully unlock the parallel potential of dLLMs, as highly parallel decoding consistently leads to degraded performance.

This paper focuses on training dLLMs to unleash their potential for parallel decoding. We identify the core bottleneck as their sequential certainty convergence. Although dLLMs predict all masked tokens in parallel at each step, the certainty of these predictions still converges in a left-to-right sequential order. This sequential propagation of certainty prevents the model from reliably determining

---

[*]Correspoding Author

**Original LLaDA-8B-Instruct :**                         **GSM8K Acc: 75.7%**    **Avg Step: 256**

```
Step 0: [Prompt]  To  determine  how  many
Step 1: [Prompt]  To  determine  how  many  kilometers
Step 2: [Prompt]  To  determine  how  many  kilometers       can
Step 3: [Prompt]  To  determine  how  many  kilometers       can  run
Step 4: [Prompt]  To  determine  how  many  kilometers       can  run  in
Step 5: [Prompt]  To  determine  how  many  kilometers  Lily can  run  in
Step 6: [Prompt]  To  determine  how  many  kilometers  Lily can  run  in            hours,
Step 7: [Prompt]  To  determine  how  many  kilometers  Lily can  run  in    eight  hours,
Step 8: [Prompt]  To  determine  how  many  kilometers  Lily can  run  in    eight  hours,  we
Step 9: [Prompt]  To  determine  how  many  kilometers  Lily can  run  in    eight  hours,  we  need
```

**8.5x Speedup**

**dParallel (Ours) :**                              **GSM8K Acc: 76.1%**    **Avg Step: 30**

```
Step 0: [Prompt]  To  determine  how  many
Step 1: [Prompt]  To  determine  how  many  kilometers  Lily  can  run  in  eight  hours,  we  need
```

Figure 1: Our method achieves highly parallel decoding. Compared to the original LLaDA Model, dParallel decodes over 8 tokens per step on GSM8K while preserving the accuracy.

multiple tokens simultaneously, forming the key bottleneck to highly parallel decoding. Employing naive teacher forcing or diffusion forcing (Chen et al., 2024) training is insufficient to resolve this issue, as they solely focus on trajectory alignment. Consequently, a new training paradigm centered on predictive certainty itself is needed for dLLMs to further unlock parallelism.

Building on this insight, we present certainty-forcing distillation, a simple and effective training strategy that directly leverages token certainty as a training signal. The core idea is to convert dLLM's inherently sequential certainty propagation into a more parallel convergence process. Concretely, we guide a pretrained dLLM to self-distill along its original semi-autoregressive decoding trajectory to maintain trajectory consistency, while simultaneously minimizing its predictive entropy over correctly predicted masked tokens to enforce high certainty. Certainty-forcing enables more tokens to reach high certainty in parallel at each step, thereby significantly extending the boundary of parallel decoding in dLLMs.

We evaluate the effectiveness of our method on two representative open-source dLLMs: LLaDA, a native dLLM trained from scratch, and Dream, a dLLM initialized from an autoregressive LLM. Comprehensive experiments across multiple benchmarks demonstrate that our approach significantly reduces the number of decoding steps in dLLMs, while maintaining comparable performance. For instance, when applied to the LLaDA-8B-Instruct model, our approach achieves an 88% reduction in decoding steps on GSM8K (Cobbe et al., 2021), yielding an $8.5\times$ speedup without sacrificing accuracy (Fig.1). On MBPP (Austin et al., 2021b), it further reduces decoding steps by 91%, delivering a $10.5\times$ acceleration while maintaining performance. Furthermore, the training process of our method is highly efficient and low-cost. Leveraging Low-Rank Adaptation (LoRA) (Hu et al., 2022), the training can be completed in just 10 hours on only eight A5000 GPUs with 24 GB memory each.

In conclusion, we present dParallel, a learnable approach that unleashes the potential of parallel decoding in dLLMs, drastically reducing the number of decoding steps. Our analysis identifies the core bottleneck as the sequential convergence of certainty across masked tokens. To address this, we introduce a certainty-forcing distillation strategy that ensures consistency with the original generation trajectory while encouraging masked tokens to attain high certainty faster and more in parallel. Extensive experiments demonstrate the effectiveness of our method. This work establishes a new baseline and provides a foundation for future research on few-step and parallel dLLMs.

## 2 RELATED WORKS

**Diffusion Language Models.** In recent years, diffusion models (Ho et al., 2020; Song et al., 2020) have established dominance in the field of visual generation (Rombach et al., 2022; Podell et al., 2023; Ruiz et al., 2023; Zhang et al., 2023). However, their application to text generation remains highly challenging. Masked diffusion models (Shi et al., 2024; Austin et al., 2021a; Sahoo et al., 2024; Zheng et al., 2024; Lou et al., 2023)have emerged as a promising approach, modeling language in the discrete space by predicting masked tokens, thereby offering the potential for fast and parallel

decoding. Building upon this idea, two representative dLLMs, LLaDA (Nie et al., 2025) and Dream (Ye et al., 2025), have recently attracted significant attention from the community, demonstrating that dLLMs can achieve performance comparable to autoregressive LLMs at the billion-parameter scale. Beyond these developments, there is also growing interest in reasoning dLLMs (Zhao et al., 2025; Wang et al., 2025b; Zhu et al., 2025), multimodal dLLMs (You et al., 2025; Yu et al., 2025b; Yang et al., 2025; Li et al., 2025c), and code generation (Gong et al., 2025; Xie et al., 2025) dLLMs.

**Accelerating Diffusion Language Models.** The potential of dLLMs in inference efficiency remains largely underexplored. Recent studies have increasingly focused on accelerating the decoding process of dLLMs. Some approaches (Ma et al., 2025; Liu et al., 2025; Wu et al., 2025; Hu et al., 2025; Chen et al., 2025) aim to reduce the time cost for each decoding step by enabling caching mechanisms and employing token dropping during inference. Other works (Israel et al., 2025; Wei et al., 2025; Li et al., 2025a;b; Gwak et al., 2025; Ben-Hamu et al., 2025) focus on reducing the total number of decoding steps by designing improved sampling strategies. In addition, hybrid methods (Wang et al., 2025a; Arriola et al., 2025)have been proposed that combine the generative paradigms of dLLMs and autoregressive LLMs, training models to realize more efficient inference pipelines. SDTT Deschenaux & Gulcehre (2024) employs progressive distillation to reduce the inference steps. Further effort Xu & Yang (2025) leverages quantization techniques to construct lightweight dLLMs.

## 3 PRELIMINARIES

**Masked Diffusion Language Models (MDLMs).** Unlike AR-LLMs that predict tokens in a strict left-to-right fashion, MDLMs (Shi et al., 2024; Austin et al., 2021a; Zheng et al., 2024)formulate generation as a probabilistic process consisting of a forward *masking* corruption and a reverse *denoising* recovery. The forward process corrupts a clean sequence $x_0$ into $x_t$ at level $t \in [0, 1]$:

$$q(x_t \mid x_0) = \prod_{i=1}^{L} \Big[ (1-t)\, \delta(x_t^i = x_0^i) + t\, \delta(x_t^i = \texttt{[MASK]}) \Big]. \tag{1}$$

The reverse process is parameterized by a mask predictor $p_\theta$, which attempts to recover $x_0$ from $x_t$. At each step, the model predicts all masked tokens jointly:

$$p_\theta(x_0 \mid x_t) = \prod_{i:x_t^i = \texttt{[MASK]}} p_\theta(x_0^i \mid x_t), \tag{2}$$

The training objective, defined as the negative log-likelihood restricted to masked positions, has been shown to upper bound the model's negative log-likelihood (Ou et al., 2024):

$$\mathcal{L}(\theta) = -\mathbb{E}_{t,x_0,x_t} \left[ \frac{1}{t} \sum_{i=1}^{L} \mathbf{1}[x_t^i = \texttt{[MASK]}] \log p_\theta(x_0^i \mid x_t) \right]. \tag{3}$$

**Sampling Process.** Inference proceeds through a discretized reverse process: at each step the model predicts distributions for all masked tokens in parallel, samples provisional tokens, and then applies a dynamic remasking strategy to determine which positions remain masked for further refinement. Unlike autoregressive decoding, this procedure allows multiple tokens to be determined in parallel, thereby enabling more flexible and potentially faster generation.

## 4 METHOD

### 4.1 THE BARRIERS TO PARALLEL DECODING

Diffusion language models are designed, in principle, for highly parallel token prediction. Yet in practice, this theoretical promise breaks down. To understand this discrepancy, we analyze the certainty dynamics of token predictions in dLLMs, revealing why their potential for parallel decoding remains unrealized.

**Certainty Correlates with Prediction Accuracy.** We first establish that token-level certainty is a reliable indicator of prediction correctness. Using LLaDA-8B-Instruct on the GSM8K test set

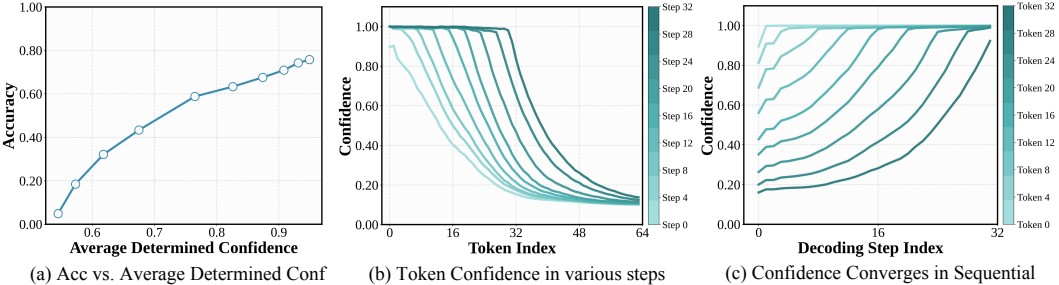

Figure 2: Empirical Studies: (a) The average confidence score exhibits a positive correlation with generation accuracy. (b) Token confidence propagates sequentially during the decoding process. (c) Convergence trajectories of confidence for different tokens.

(Cobbe et al., 2021), we adopt a remasking strategy with varying confidence thresholds and record the average determined confidence of tokens. Fig 2 (a) shows a strong positive correlation between token confidence and the generation correctness: tokens resolved at higher confidence consistently achieve higher accuracy, whereas low-confidence commitments lead to frequent errors. This result confirms that high certainty is a necessary condition for accurate generation.

**Certainty Converges to Peak Sequentially.** The high certainty is not achieved in parallel. Instead, it propagates sequentially through the sequence. At any given decoding step, the model predicts all masked tokens, but only a small subset, typically those adjacent to already known context, attain high confidence. The vast majority of tokens remain in a low-confidence regime until a new context becomes available. Once a confident token is committed, it provides a new conditioning context that allows another subset to rise in certainty at the next step.

This dynamic is illustrated in Fig 2 (b), which shows the average confidence of tokens progressing as a left-to-right propagation over decoding steps. Fig 2 (c) further confirms this at the individual token level, showing confidence trajectories that converge to high certainty in a staggered, sequential order. Together, these findings reveal that high certainty does not emerge in parallel but unfolds sequentially through iterative context enrichment.

**The Fundamental Bottleneck.** The key bottleneck is the sequential convergence of certainty. While true parallelism requires committing many tokens in a single step, a dLLM gains high certainty only for a few neighboring tokens per iteration. Forcing multiple commitments too early introduces low-confidence predictions, causing cascading errors and performance degradation.

**Key to Unlocking Parallelism Potential.** The above insight illuminates a clear path forward: if we could guide the model to achieve peak confidence in parallel across multiple token positions, we could break the sequential bottleneck. However, traditional training strategies, such as teacher forcing and diffusion forcing (Chen et al., 2024), are inadequate for this purpose, as their focus on trajectory alignment overlooks the dynamics of predictive certainty. Consequently, unlocking greater parallelism in dLLMs requires a new training paradigm that directly optimizes for certainty. We therefore propose certainty-forcing disillation, a novel strategy that reshapes the model's certainty dynamics by using token certainty itself as a direct training signal.

## 4.2 CERTAINTY-FORCING DISTILLATION

We propose certainty-forcing distillation, a straightforward approach that enforces parallel certainty along the original trajectory without altering it. An overview is shown in Fig 3.

**Teacher Trajectory Generation.** Let $M_{\theta_T}$ be the teacher model (a pre-trained vanilla dLLM), and let $M_{\theta_S}$ be the student model, initialized as an identical copy. We train on a dataset $\mathcal{D} = \{X^{(i)}\}_{i=1}^{K}$, where each $X^{(i)}$ is an instruction prompt. For each prompt, the teacher $M_{\theta_T}$ generates a target response trajectory using a semi-autoregressive remasking strategy with total length $L$ and block size $L_b$, producing a sequence $Y = (y_1, y_2, \ldots, y_L)$. This sequence is partitioned into $N$ contiguous blocks $\{B_1, B_2, \ldots, B_N\}$ such that $L = N \times L_b$, where the $n$-th block is defined as $B_n = (y_{(n-1)L_b+1}, \ldots, y_{nL_b})$ for $n \in \{1, \ldots, N\}$.

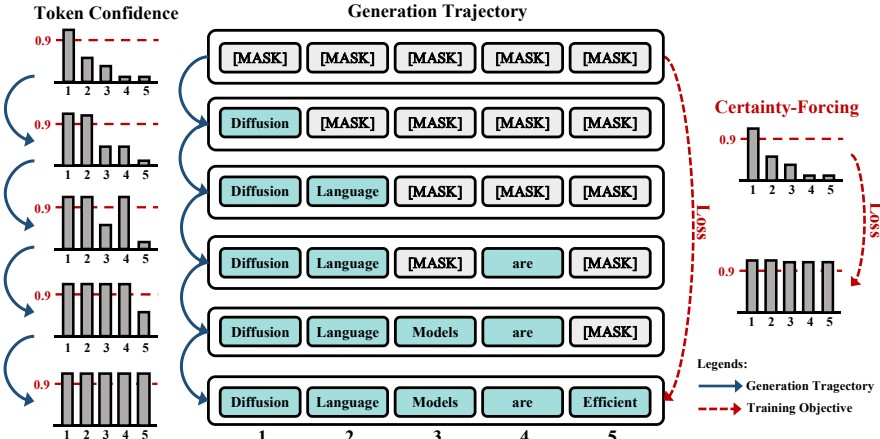

Figure 3: Overview of proposed certainty-forcing distillation. The dLLM is self-distilled along its original generation trajectory, ensuring consistency with the trajectory throughout training while encouraging token certainty to converge faster in parallel rather than sequentially.

**Semi-Autoregressive Forward Masking.** To simulate the trajectory generation process for training, we perturb the clean trajectory $Y$ and create a noisy input sequence $\tilde{Y}$ by applying a semi-autoregressive structural masking scheme. We first uniformly sample a block index $n \sim \{0, \ldots, N-1\}$. The sequence is then divided into three distinct parts based on this index: (1) Context Blocks ($i \leq nL_b$): Tokens within the first $n$ blocks remain unmasked, serving as the model's context. (2) Active Block ($nL_b < i \leq (n+1)L_b$): This is the block currently being generated, where its tokens are randomly replaced by the token [MASK] with masking probability $p_m = q$. (3) Future Blocks ($i > (n+1)L_b$): All tokens in subsequent blocks are fully masked, as they have not yet been generated. This procedure yields a noisy token $\tilde{y}_i$ at each position $i$, defined as:

$$\tilde{y}_i = \begin{cases} y_i, & \text{if } i \leq nL_b \quad \text{(Context)} \\ \begin{cases} y_i & \text{with probability } 1-q \\ \text{[MASK]} & \text{with probability } q \end{cases} & \text{if } nL_b < i \leq (n+1)L_b \quad \text{(Active Block)} \\ \text{[MASK]}, & \text{if } i > (n+1)L_b \quad \text{(Future)} \end{cases} \quad (4)$$

The resulting sequence $\tilde{Y}$ simulates an intermediate state in the semi-autoregressive generative process, where the model is predicting the $(n+1)$-th block given the context of the first $n$ blocks.

**Training Objective.** Our objective differs from standard dLLM pre-training, which typically aims to predict all masked tokens across the sequence. Instead, we restrict the learning signal to the masked tokens within the active block $B_{n+1}$. Our training objective is for the student model not only to replicate the target sampling trajectory within the active block but also to parallel achieve maximal certainty in its predictions.

To enforce consistency between the student model's generated trajectory and that of the teacher, we apply standard Cross-Entropy (CE) loss on the masked tokens of the active block, denoted as $\mathcal{M}_a$:

$$\mathcal{L}_{\text{Consistency}} = -\frac{1}{|\mathcal{M}_a|} \sum_{i \in \mathcal{M}_a} \log p_\theta(y_i \mid \tilde{Y}), \quad (5)$$

where $p_\theta(y_i \mid \tilde{Y})$ denotes the probability assigned by the student model to the correct token $y_i$ at position $i$, conditioned on the noisy input sequence $\tilde{Y}$. However, conventional CE loss is insufficient for our certainty-maximizing target. It focuses solely on correctness, and once the correct token is predicted, the gradient quickly vanishes, offering no incentive to further increase confidence.

To explicitly encourage highly confident predictions, we introduce a term that directly minimizes the entropy of the model's output distribution, incorporating a temperature parameter $T$. This loss is applied only to the masked tokens in the active block that the student model already predicts

---

**Algorithm 1** Certainty-Forcing Distillation (CFD)

---

**Require:** Teacher $M_{\theta_T}$, student $M_{\theta_S}$; target trajectory set $\mathcal{D} = \{Y^{(i)}\}_{i=1}^{K}$; temperature $T > 0$;
  weight $\beta \geq 0$; optimizer $\mathcal{O}$; token length $L$; block length $L_b$; mask ratio $q \in (0, 1]$.
**Notation:** $H(p) = -\sum_{v \in \mathcal{V}} p(v) \log p(v)$
1: **for** $j = 1 \ldots Iteration$ **do**
2:   Sample $Y \sim \mathcal{D}$, $n \sim \{0, 1, \ldots, L/L_b - 1\}$;
3:   $(\tilde{Y}, \mathcal{M}_a) \leftarrow$ SEMI-AR-FOWARDMASKING$(Y, q, n, L, L_b)$;
4:   $z \leftarrow M_{\theta_S}(\tilde{Y})$;
5:   $p_i(v) \leftarrow \text{softmax}(z_i)_v, \quad p_i^{(T)}(v) \leftarrow \text{softmax}(z_i/T)_v, \quad \forall v \in \mathcal{V}$
6:   $\mathcal{L}_{\text{Consistency}} \leftarrow -|\mathcal{M}_a|^{-1} \sum_{i \in \mathcal{M}_a} \log p_i(y_i)$;
7:   $\mathcal{M}_c \leftarrow \{ i \in \mathcal{M}_a \mid \arg\max_{v \in \mathcal{V}} p_i(v) = y_i \}$;
8:   $\mathcal{L}_{\text{Certainty}} \leftarrow \mathbf{1}[|\mathcal{M}_c| > 0] \cdot |\mathcal{M}_c|^{-1} \sum_{i \in \mathcal{M}_c} H(p_i^{(T)})$;
9:   $\mathcal{L}_{\text{CFD}} \leftarrow \mathcal{L}_{\text{Consistency}} + \beta \mathcal{L}_{\text{Certainty}}$;
10:   $\theta_S \leftarrow \mathcal{O}(\theta_S, \nabla_{\theta_S} \mathcal{L}_{\text{CFD}})$;
11: **end for**

---

correctly. Formally, we define the set of correctly predicted tokens as

$$\mathcal{M}_c = \left\{ i \in \mathcal{M}_a \;\middle|\; \arg\max_{v \in \mathcal{V}} p_\theta(v \mid \tilde{Y}) = y_i \right\}, \tag{6}$$

where $\mathcal{V}$ denotes the vocabulary. The certainty-forcing loss is then defined as the average entropy of the predictive distributions for these tokens:

$$\mathcal{L}_{\text{Certainty}} = \frac{1}{|\mathcal{M}_c|} \sum_{i \in \mathcal{M}_c} \left( -\sum_{v \in \mathcal{V}} p_\theta(v \mid \tilde{Y}; T) \log p_\theta(v \mid \tilde{Y}; T) \right), \tag{7}$$

where $p_\theta(v \mid \tilde{Y}; T)$ denotes the temperature-scaled softmax distribution. Minimizing this term encourages the student model to generate sharper, higher-certainty distributions over the correct tokens, where $T$ controls the strength of the certainty enforcement.

The overall training objective is a combination of consistency loss and the certainty-forcing loss:

$$\mathcal{L}_{\text{CFD}} = \mathcal{L}_{\text{Consistency}} + \beta \mathcal{L}_{\text{Certainty}}, \tag{8}$$

where $\beta$ is a hyperparameter balancing the objective of matching the teacher's trajectory with the objective of enforcing high certainty. We find that this simple distillation strategy significantly accelerates the parallel convergence of certainty in dLLMs, thereby unlocking their inherent potential for parallel decoding. The overall training pipeline is summarized in Algorithm 1

## 5 EXPERIMENTS

### 5.1 EXPERIMENTAL SETUP

**Implementation Details.** We evaluate the effectiveness of our method on two representative open-source dLLMs: LLaDA-8B-Instruct (Nie et al., 2025) and Dream-7B-Instruct (Ye et al., 2025). The training is conducted using the LoRA technique (Hu et al., 2022). For semi-autoregressive masking, we set the block length to $L_b = 32$ for LLaDA and $L_b = 256$ for Dream, with a fixed masking ratio of 50%. The certainty loss is applied with a temperature of $T = 0.5$. Full training configurations are provided in the appendix. During inference, our models adopt an entropy-threshold semi-autoregressive remasking strategy , which is inherently consistent with our training objective.

**Training Data.** As a self-distillation approach, we use prompts from publicly available training datasets and let the pretrained model generate its own responses as training data. For LLaDA-8B-Instruct, we sample prompts from the GSM8K (Cobbe et al., 2021), PRM12K (Lightman et al., 2023) training set, and part of the Numina-Math dataset (Li et al., 2024). We generate target trajectories using a semi-autoregressive strategy with a sequence length of 256 and block length of 32. We further filter out responses containing incorrect answers and finally get about 92k samples. For

Table 1: Evaluation results on LLaDA-8B-Instruct. For all methods, we adopt a semi-autoregressive remasking strategy with a total sequence length of 256 and a block length of 32. For our approach, the entropy threshold is set to either 0.45 or 0.5 for different tasks.

| Benchmark | Method | #Steps ↓ | Latency↓ | Speedup ↑ | Accuracy↑ |
|---|---|---|---|---|---|
| **GSM8K -CoT** (0-shot) | LLaDA-8B-Instruct | 256 | 18.6s | 1.0× | 75.7% |
| | Dual-Cache | 256 | 9.7s | 1.9× | 72.9% |
| | Few-step Decoding | 64 | 4.7s | 4.0× | 68.6% |
| | Conf-threshold Decoding | 72 | 5.2s | 3.6× | 75.5% |
| | Consistency Distillation | 64 | 4.7s | 4.0× | 69.9% |
| | **dParallel (Ours)** | **30** | **2.2s** | **8.5×** | 76.1% |
| **MATH** (4-shot) | LLaDA-8B-Instruct | 256 | 50.9s | 1.0× | 33.5% |
| | Dual-Cache | 256 | 11.3s | 4.5× | 32.6% |
| | Few-step Decoding | 64 | 12.7s | 4.0× | 26.3% |
| | Conf-threshold Decoding | 97 | 17.6s | 2.9× | 33.2% |
| | Consistency Distillation | 64 | 12.7s | 4.0× | 28.0% |
| | **dParallel (Ours)** | **46** | **8.9s** | **5.7×** | 31.5% |
| **HumanEval** (0-shot) | LLaDA-8B-Instruct | 256 | 23.5s | 1.0× | 38.4% |
| | Dual-Cache | 256 | 9.8s | 2.4× | 34.1% |
| | Few-step Decoding | 64 | 5.9s | 4.0× | 19.5% |
| | Conf-threshold Decoding | 77 | 6.7s | 3.5× | 37.2% |
| | Consistency Distillation | 64 | 5.9s | 4.0× | 19.5% |
| | **dParallel (Ours)** | **33** | **2.9s** | **8.2×** | 40.2% |
| **MBPP** (3-shot) | LLaDA-8B-Instruct | 256 | 50.1s | 1.0× | 42.4% |
| | Dual-Cache | 256 | 10.7s | 4.7× | 39.8% |
| | Few-step Decoding | 64 | 12.5s | 4.0× | 19.6% |
| | Conf-threshold Decoding | 68 | 12.8s | 3.9× | 41.6% |
| | Consistency Distillation | 64 | 12.5s | 4.0× | 25.0% |
| | **dParallel (Ours)** | **24** | **4.8s** | **10.5×** | 40.8% |

Dream-7B-Instruct, we adopt the same trajectory generation strategy, and additionally generate code data using prompts from a subset of the AceCode dataset (about 10k) (Zeng et al., 2025). Importantly, all training tokens are generated by the model itself, without introducing any external data as targets.

**Evaluation Details.** We evaluate our models across multiple benchmarks, including two mathematics datasets (GSM8K and MATH (Lewkowycz et al., 2022)) and two code generation datasets (HumanEval (Chen et al., 2021) and MBPP (Austin et al., 2021b)). For GSM8K, we append a chain-of-thought (CoT) prompt (Wei et al., 2022) after each question. We report accuracy, the average number of decoding steps, latency, and speedup ratio to provide a comprehensive evaluation. All efficiency evaluations are conducted on NVIDIA RTX 6000 Ada GPUs.

**Baselines.** We evaluate the original dLLM under its official default inference setting, and further compare our approach with four baselines that seek to accelerate the generation: (1) Dual-Cache: enable KV-cache on both prefix tokens and suffix tokens (Wu et al., 2025). (2) Few-step Decoding: reducing the number of decoding steps used by the original dLLM. (3) Conf-threshold Decoding: apply adaptive remasking based on the model's confidence in predicting masked tokens (Wu et al., 2025; Yu et al., 2025b), with the confidence threshold set as 0.90 or 0.95 depending on the task. (4) Consistency Distillation: training the dLLM to predict all remaining masked tokens from intermediate state along its own generation trajectory (Luo et al., 2023). The training data and LoRA configuration are the same as our method.

## 5.2 MAIN RESULTS

**Results on the Native LLaDA Model.** As shown in Table 1, directly reducing the decoding steps of the original model leads to a substantial drop in performance. Consistency distillation has only a marginal effect on LLaDA, offering a slight improvement over the original model under the same number of steps. The confidence-threshold remasking strategy preserves accuracy, but its parallelism is limited, averaging only 3–4 tokens decoded per step. In contrast, our method significantly pushes the boundaries of parallel inference in dLLMs, achieving more than 8 tokens decoded per

Table 2: Evaluation results on Dream-8B-Instruct. The original model uses the official inference setting with a sequence length of 256. Other methods adopt semi-autoregressive remasking with the same length and a block size of 32. The entropy threshold for our method is set to either 0.45 or 0.5.

| Benchmark | Method | #Steps ↓ | Latency↓ | Speedup ↑ | Accuracy↑ |
|---|---|---|---|---|---|
| GSM8K -CoT (0-shot) | Dream-7B-Instruct | 256 | 17.2s | 1.0× | 82.9% |
| | Dual-Cache | 256 | 8.2s | 2.1× | 79.5% |
| | Few-step Decoding | 64 | 4.3s | 4.0× | 59.0% |
| | Conf-threshold Decoding | 61 | 4.0s | 4.3× | 81.9% |
| | Consistency Distillation | 64 | 4.3s | 4.0× | 75.6% |
| | **dParallel (Ours)** | **39** | **2.5s** | **6.9×** | 82.1% |
| MATH (0-shot) | Dream-7B-Instruct | 256 | 17.5s | 1.0× | 39.5% |
| | Dual-Cache | 256 | 8.2s | 2.1× | 38.8% |
| | Few-step Decoding | 64 | 4.4s | 4.0× | 16.7% |
| | Conf-threshold Decoding | 93 | 6.1s | 2.9× | 38.9% |
| | Consistency Distillation | 64 | 4.4s | 4.0× | 29.6% |
| | **dParallel (Ours)** | **63** | **4.1s** | **4.2×** | 38.3% |
| HumanEval -Instruct (0-shot) | Dream-7B-Instruct | 256 | 25.9s | 1.0× | 52.4% |
| | Dual-Cache | 256 | 8.4s | 3.1× | 47.0% |
| | Few-step Decoding | 64 | 6.5s | 4.0× | 16.5% |
| | Conf-threshold Decoding | 71 | 7.3s | 3.5× | 53.1% |
| | Consistency Distillation | 64 | 6.4s | 4.0× | 34.2% |
| | **dParallel (Ours)** | **37** | **3.8s** | **6.9×** | 54.3% |
| MBPP -Instruct (0-shot) | Dream-7B-Instruct | 256 | 19.8s | 1.0× | 58.8% |
| | Dual-Cache | 256 | 8.9s | 2.2× | 52.8% |
| | Few-step Decoding | 64 | 5.0s | 4.0× | 25.0% |
| | Conf-threshold Decoding | 43 | 3.3s | 5.9× | 56.4% |
| | Consistency Distillation | 64 | 5.0s | 4.0× | 37.4% |
| | **dParallel (Ours)** | **29** | **2.2s** | **8.8×** | 56.2% |

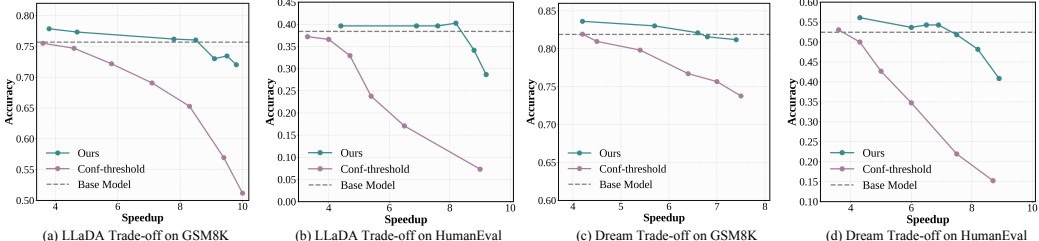

Figure 4: Comparison of speed–accuracy trade-off curves between confidence-threshold decoding and our method. (a) and (b) show results on the LLaDA model for GSM8K and HumanEval, respectively. (c) and (d) present results on the Dream model for GSM8K and HumanEval benchmarks.

step on average while still maintaining performance. Notably, for LLaDA, we trained using only prompts from mathematical tasks, yet the model still exhibited a remarkable improvement in parallel decoding ability on code tasks.

**Results on the AR-initialized Dream Model.** As shown in Table 2, our method also demonstrates superior performance on the Dream model, which is initialized from an AR-LLM. Compared to other approaches designed to reduce the number of decoding steps, dParallel achieves a substantially higher speedup while maintaining accuracy, thereby greatly enhancing decoding parallelism. It is worth noting that we observed a risk of degeneration toward the original AR LLM when training Dream with semi-autoregressive masking. To avoid this issue, we employed standard random masking over the entire sequence instead. Consequently, the acceleration gains of our method on Dream are slightly lower than those observed on LLaDA.

**Superior Efficiency–Performance Trade-off.** In Fig 4, we compare our method against the original model with confidence-threshold decoding in terms of the efficiency–performance trade-off curve. Our approach achieves a substantially better trade-off. On LLaDA with GSM8K, at the same $9.4\times$ speedup, our method attains $16.5\%$ higher accuracy than confidence-threshold decoding. On HumanEval, at the same $9.3\times$ speedup, our method improves accuracy by $21.3\%$. Results on Dream

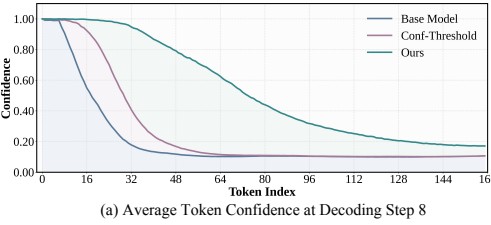 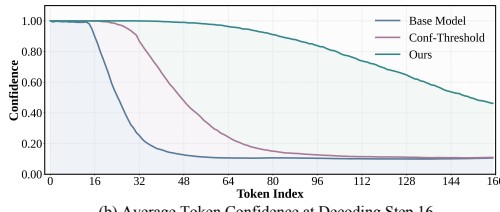

(a) Average Token Confidence at Decoding Step 8     (b) Average Token Confidence at Decoding Step 16

Figure 5: Average token confidence at the 8th and 16th decoding steps for LLaDA-8B-Instruct Model on GSM8K. The proposed certainty-forcing strategy reshapes the original sequential certainty convergence into a faster and more parallel convergence process.

Table 3: Ablation study on different training strategies of our method using the LLaDA model.

| Consistency Loss | Certainty Loss | Semi-AR Masking | GSM8K-CoT (0-shot) | | | HumanEval (0-shot) | | |
|:---:|:---:|:---:|:---:|:---:|:---:|:---:|:---:|:---:|
| | | | #Steps ↓ | Speed ↑ | Acc ↑ | #Steps ↓ | Speed ↑ | Acc ↑ |
| ✓ | | ✓ | 53 | 4.5× | 73.5% | 71 | 3.6× | 36.0% |
| | ✓ | ✓ | 23 | 10.4× | 57.8% | 28 | 9.8× | 30.5% |
| ✓ | ✓ | | 44 | 5.5× | 73.3% | 61 | 4.3× | 32.9% |
| ✓ | ✓ | ✓ | 30 | 8.5× | 76.1% | 33 | 8.2× | 40.2% |

exhibit a similar curve. These findings strongly demonstrate that our method effectively broadens the boundary of parallel decoding in diffusion language models.

**Faster and Parallel Certainty Convergence.** As illustrated in Fig 5, the original dLLM exhibits a sequential convergence of token certainty, where each step produces high confidence only for a small set of neighboring tokens, while the majority remain in a low-confidence range. Confidence-based decoding can extend the boundary of token certainty but still follows a sequential propagation pattern. In contrast, our dParallel, trained with certainty-forcing distillation, transforms this process into a significantly faster and more parallel convergence of certainty. Such parallel convergence further unlocks the potential of dLLMs for highly efficient parallel decoding.

## 5.3 ABLATION STUDY

**Ablation Study on Training Strategy.** We conducted an ablation study to validate the effectiveness of our proposed certainty-forcing distillation, with the results shown in Table 3. When the certainty-forcing loss is removed, the remaining consistency loss is insufficient to alter the sequential convergence pattern of the dLLM, resulting in speed and performance similar to the baseline model. Conversely, applying only the certainty loss without enforcing trajectory consistency achieves high decoding speed but leads to a sharp performance drop. Finally, our use of semi-autoregressive forward masking effectively aligns the trajectory generation process with self-distillation, yielding superior efficiency and performance. These results collectively demonstrate that each component in the training process is essential.

**Ablation Study on Masking Ratio.** We conduct an ablation study to determine the optimal masking ratio, training LLaDA for one epoch with various settings as shown in Table 4. We find that a fixed masking ratio of 50% yields the best performance, offering significant acceleration while preserving accuracy. In contrast, both higher and lower fixed ratios, as well as random ratios, lead to a noticeable accuracy degradation. This suggests that a 50% ratio creates an optimal trade-off between the training signals for consistency and certainty by balancing masked and unmasked tokens.

Table 4: Performance of our method applied to LLaDA-8B-Instruct on GSM8K with different masking ratios used in the forward process during training.

| Masking Ratio | #Steps ↓ | Speed ↑ | Acc ↑ |
|:---:|:---:|:---:|:---:|
| **Random** | 36 | 7.1x | 72.4% |
| **25%** | 35 | 7.3x | 69.9% |
| **75%** | 38 | 6.7x | 73.7% |
| **100%** | 31 | 8.3x | 71.1% |
| **50%** | 36 | 7.1x | **76.3%** |

Importantly, training with this fixed ratio does not impair the model's ability to handle variable ratios during inference.

**Prompt:** Lily can run 12 kilometers per hour for 4 hours. After that, she runs 6 kilometers per hour. How many kilometers can she run in 8 hours? Please reason step by step, and put your final answer within \boxed{}.

| | |
|---|---|
| **LLaDA-8B-Instruct**                **# 256 Decoding Steps** | **dParallel (Ours)**                **# 28 Decoding Steps** |

To determine how many kilometers Lily can run in 8 hours, we need to break down the problem into two parts: the distance she runs in the first 4 hours and the distance she runs in the next 4 hours.

First, let's calculate the distance she runs in the first 4 hours:
- She runs at a speed of 12 kilometers per hour for 4 hours.
- Distance = Speed × Time
- Distance = 12 km/h × 4 hours = 48 km

Next, let's calculate the distance she runs in the next 4 hours:
- She runs at a speed of 6 kilometers per hour for 4 hours.
- Distance = Speed × Time
- Distance = 6 km/h × 4 hours = 24 km

Now, we add the distances from both parts to find the total distance she can run in 8 hours:
- Total distance = Distance in the first 4 hours + Distance in the next 4 hours
- Total distance = 48 km + 24 km = 72 km

Therefore, the total distance Lily can run in 8 hours is \boxed{72} kilometers.

---

To determine how many kilometers Lily can run in 8 hours, we need to calculate the distance she covers in each segment of her run and then sum these distances.

1. **Calculate the distance run in the first 4 hours:**
   - Speed: 12 kilometers per hour
   - Time: 4 hours
   - Distance: Speed × Time = 12 km/h × 4 hours = 48 kilometers

2. **Calculate the distance run in the next 4 hours:**
   - Speed: 6 kilometers per hour
   - Time: 4 hours
   - Distance = Speed × Time = 6 km/h × 4 hours = 24 kilometers

3. **Sum the distances from both segments:**
   - Total distance = Distance in the first 4 hours + Distance in the next 4 hours
   - Total distance = 48 km + 24 km = 72 kilometers

Therefore, the total distance Lily can run in 8 hours is \boxed{72}.

Figure 6: Case study on LLaDA-8B-Instruct Model with chain-of-thought reasoning problem.

# 6   CASE STUDY

In Figure 6, we present a case study of our dParallel model. When tackling the same Chain-of-Thought (CoT) math reasoning problem, dParallel achieves an answer quality using only 28 decoding steps that is on par with the original LLaDA-8B-Instruct model requiring 256 steps. Furthermore, because dParallel selects the reasoning path with the lowest information entropy, the total number of tokens required to answer the question is slightly reduced. This case study clearly demonstrates the effectiveness of our approach in enhancing the decoding parallelism of dLLMs.

# 7   CONCLUSION

In this paper, we present dParallel, a simple yet effective method that unleashes the parallel decoding potential of dLLMs. At the core of our approach is certainty-forcing distillation, a novel training strategy that maintains trajectory consistency while compelling high-certainty predictions, thus overcoming the sequential certainty propagation issue. Extensive experiments across various benchmarks validate the effectiveness of our method. Our work establishes a new baseline for parallel decoding in dLLMs and explores a new avenue for dLLM training paradigms.

## ACKNOWLEDGEMENT

This project is supported by the Ministry of Education, Singapore, under its Academic Research Fund Tier 2 (Award Number: MOE-T2EP20122-0006).

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

APPENDIX

## A  MORE IMPLEMENTATION DETAILS

In Table 5, we present the training configuration used for the certainty-forcing distillation process. For data generated by LLaDA-8B-Instruct (Nie et al., 2025) and LLaDA-1.5 (Zhu et al., 2025), we standardized sequence lengths by padding or truncating with the end-of-sequence token to a fixed length of 384 tokens. In contrast, for Dream-7B-Instruct (Ye et al., 2025), we preserved the original response length of 256 tokens per sample without modification. Additionally, we set the balance weight $\beta = 2$ for all training. We also used a complementary mask training strategy to improve token utilization.

Our training was conducted on two NVIDIA H100 GPUs, with a per-GPU mini-batch size of 1 and a gradient accumulation step of 32, resulting in an effective global batch size of 64. Notably, despite the relatively large model sizes, the adoption of parameter-efficient fine-tuning (PEFT) (Hu et al., 2022) and the use of shorter sequence lengths kept the memory footprint remarkably low. The entire training process required only 23 GB of GPU memory, meaning that it can be efficiently reproduced even on multiple consumer-grade GPUs with 24 GB of memory each. This efficiency highlights the practicality of our approach, as it enables large-scale distillation training to be carried out on widely accessible hardware rather than being restricted to specialized high-memory accelerators.

Table 5: The training configuration for certainty-forcing distillation across three base models.

| Base Model | LoRA Rank | LoRA Alpha | Learning Rate | Lr-Schedule | Batchsize | Epoch |
|---|---|---|---|---|---|---|
| LLaDA-8B-Instrcut | 32 | 32 | 2e-5 | constant | 64 | 6 |
| LLaDA-1.5 | 128 | 128 | 2e-5 | constant | 64 | 4 |
| Dream-7B-Instrcut | 16 | 16 | 2e-5 | cosine | 64 | 3 |

Table 6: Evaluation results on LLaDA-1.5 Model across four benchmarks.

| Benchmark | Method | #Steps ↓ | Latency↓ | Speedup ↑ | Accuracy↑ |
|---|---|---|---|---|---|
| **GSM8K** | LLaDA-8B-Instruct | 256 | 19.1s | 1.0× | 76.0% |
| (0-shot) | **dParallel (Ours)** | **30** | **2.3s** | **8.5×** | 76.3% |
| **MATH** | LLaDA-8B-Instruct | 256 | 50.0s | 1.0× | 34.0% |
| (4-shot) | **dParallel (Ours)** | **45** | **8.7s** | **5.7×** | 32.1% |
| **HumanEval** | LLaDA-8B-Instruct | 256 | 22.0s | 1.0× | 41.5% |
| (0-shot) | **dParallel (Ours)** | **46** | **4.0s** | **5.6×** | 40.2% |
| **MBPP** | LLaDA-8B-Instruct | 256 | 49.0s | 1.0× | 43.2% |
| (3-shot) | **dParallel (Ours)** | **26** | **5.1s** | **9.8×** | 41.6% |

## B  MORE EXPERIMENTAL RESULTS

In Table 6, we report the performance of applying our method to the LLaDA-1.5 model. Extensive evaluations across four standard benchmarks demonstrate the strong effectiveness of our approach on this reinforcement learning based model. Specifically, we reduce the original 256 decoding steps required by the baseline model to only 26–46 steps. This dramatic compression of the decoding steps delivers substantial acceleration in generation speed, while at the same time preserving accuracy and reliability across tasks.

In Figure 7, we present the average token confidence of the LLaDA-8B-Instruct model on GSM8K, measured across the first 160 positions over the initial 16 decoding steps. The results reveal that the original dLLM exhibits a clear sequential convergence of token certainty: each step yields high confidence for only a narrow band of neighboring tokens, while the majority remain in a low-confidence range. Although confidence-based decoding can extend the certainty frontier, it still follows this sequential propagation pattern. By contrast, our proposed dParallel, trained with certainty-forcing

distillation, reshapes this process into a substantially faster and more parallel convergence of certainty. This parallel convergence further unlocks the efficiency potential of dLLMs, enabling highly parallel decoding.

## C  CASE STUDY

We also present additional case studies in Figure 8, Figure 9, and Figure 10. Our dParallel achieves significantly reduced decoding steps while maintaining the generation quality.

## D  LIMITATIONS AND FUTURE WORK

The primary limitation of our method is its reliance on the performance of the pretrained dLLM. While our approach achieves substantial gains in inference efficiency by unleashing the potential of parallel decoding and maintains strong accuracy, it cannot significantly improve the performance if the base model itself is weak.

As a next step, we plan to extend our certainty-forcing strategy to the pretraining stage of dLLMs and substantially scale up the training data to explore the performance boundary of our approach. Currently, we have only used a relatively small dataset of around 10k math problems. We believe that by dramatically increasing both the size and diversity of the training data, our method can yield further improvements: not only activating highly parallel decoding, but also enhancing overall model performance and demonstrating stronger generalization.

## E  ETHICS STATEMENT

This work adheres to the ICLR Code of Ethics. Our study does not involve human subjects or sensitive personal data. All datasets used are publicly available and properly licensed. While our method focuses on improving the efficiency of diffusion language models, we recognize potential risks of misuse in harmful applications and encourage responsible use aligned with ethical and legal standards.

## F  REPRODUCIBILITY STATEMENT

We have made significant efforts to ensure the reproducibility of our work. Detailed descriptions of model architectures, training objectives, and experimental settings are provided in the main text and Appendix. All datasets used are publicly available, and their preprocessing steps are documented in the main paper and appendix. Additionally, we include pseudocode and implementation details to facilitate replication, and source code is provided in the supplementary materials.

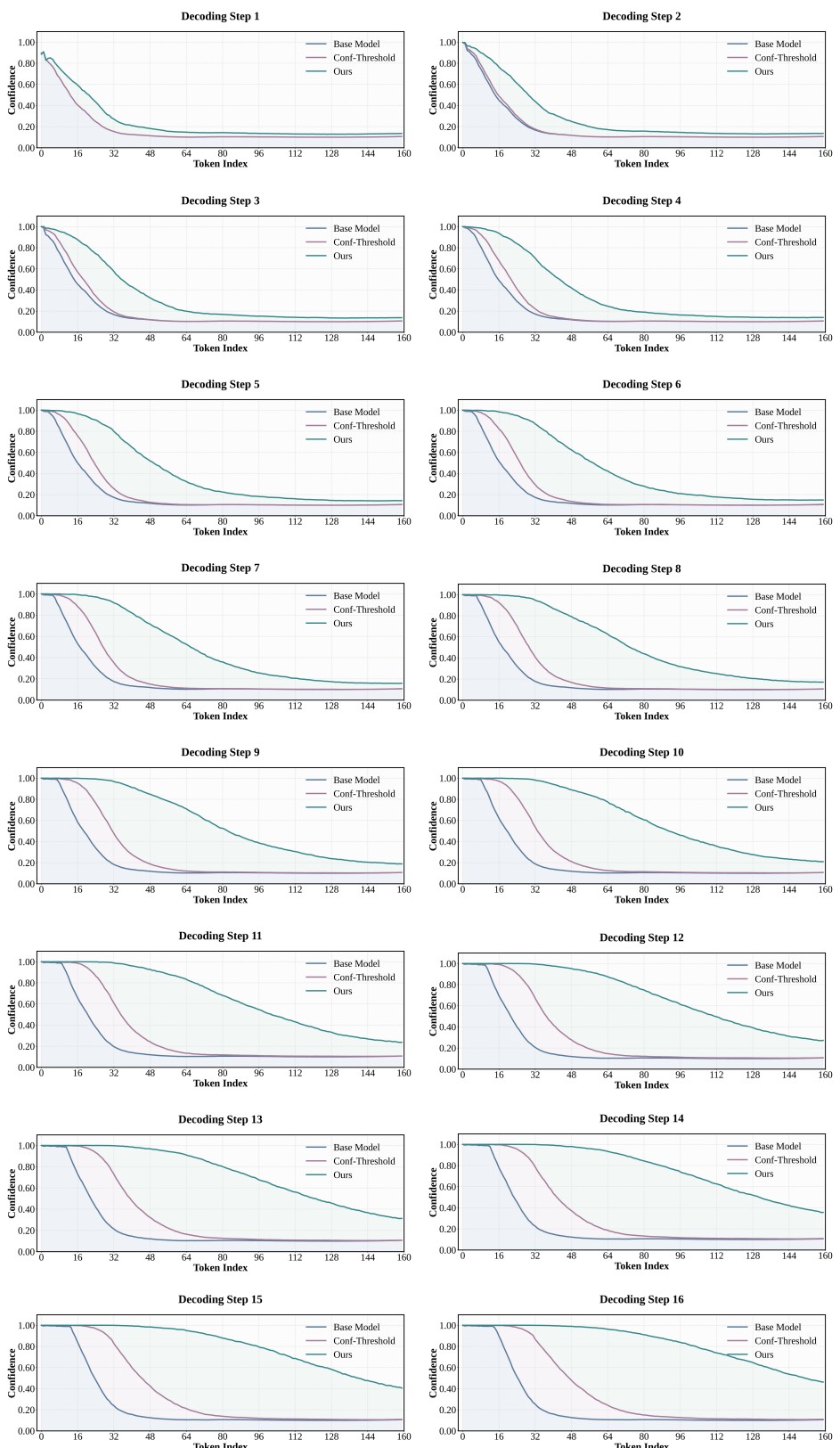

Figure 7: Average token confidence over the first 160 tokens across the first 16 decoding steps of the LLaDA-8B-Instruct model on GSM8K. Our certainty-forcing strategy transforms the sequential certainty convergence of the baseline into a faster and more parallel convergence process.

**Prompt:** Lily can run 12 kilometers per hour for 4 hours. After that, she runs 6 kilometers per hour. How many kilometers can she run in 8 hours? Please reason step by step, and put your final answer within \boxed{}.

**LLaDA-8B-Instruct**                    **# 256 Decoding Steps**

To determine how many kilometers Lily can run in 8 hours, we need to break down the problem into two parts: the distance she runs in the first 4 hours and the distance she runs in the next 4 hours.

First, let's calculate the distance she runs in the first 4 hours:
- She runs at a speed of 12 kilometers per hour for 4 hours.
- Distance = Speed × Time
- Distance = 12 km/h × 4 hours = 48 km

Next, let's calculate the distance she runs in the next 4 hours:
- She runs at a speed of 6 kilometers per hour for 4 hours.
- Distance = Speed × Time
- Distance = 6 km/h × 4 hours = 24 km

Now, we add the distances from both parts to find the total distance she can run in 8 hours:
- Total distance = Distance in the first 4 hours + Distance in the next 4 hours
- Total distance = 48 km + 24 km = 72 km

Therefore, the total distance Lily can run in 8 hours is **\boxed{72}** kilometers.

**dParallel (Ours)**                    **# 28 Decoding Steps**

To determine how many kilometers Lily can run in 8 hours, we need to calculate the distance she covers in each segment of her run and then sum these distances.

1. **Calculate the distance run in the first 4 hours:**
   - Speed: 12 kilometers per hour
   - Time: 4 hours
   - Distance: Speed × Time = 12 km/h × 4 hours = 48 kilometers

2. **Calculate the distance run in the next 4 hours:**
   - Speed: 6 kilometers per hour
   - Time: 4 hours
   - Distance = Speed × Time = 6 km/h × 4 hours = 24 kilometers

3. **Sum the distances from both segments:**
   - Total distance = Distance in the first 4 hours + Distance in the next 4 hours
   - Total distance = 48 km + 24 km = 72 kilometers

Therefore, the total distance Lily can run in 8 hours is **\boxed{72}.**

**Prompt:** Natalia sold clips to 48 of her friends in April, and then she sold half as many clips in May. How many clips did Natalia sell altogether in April and May? Please reason step by step, and put your final answer within \boxed{}.

**LLaDA-8B-Instruct**                    **# 256 Decoding Steps**

To determine the total number of clips Natalia sold in April and May, we need to follow these steps:

1. Identify the number of clips Natalia sold in April.
2. Calculate the number of clips Natalia sold in May.
3. Add the number of clips sold in April and May to find the total.

Step 1: Identify the number of clips Natalia sold in April.
Natalia sold 48 clips in April.

Step 2: Calculate the number of clips Natalia sold in May.
Natalia sold half as many clips in May as she did in April. Therefore, we calculate the number of clips sold in May as follows:
\[ \text{Number of clips sold in May} = \frac{48}{2} = 24 \]

Step 3: Add the number of clips sold in April and May to find the total.
We add the number of clips sold in April and May:
\[ \text{Total number of clips sold} = 48 + 24 = 72 \]

Thus, the total number of clips Natalia sold in April and May is **\(\boxed{72}\).**

**dParallel (Ours)**                    **# 30 Decoding Steps**

To determine the total number of clips Natalia sold in April and May, we need to follow these steps:

1. **Calculate the number of clips sold in May:**
   - Natalia sold clips to friends in May.
   - She sold half as many clips in May as she did in April.
   - Since she sold 48 clips in April:
   - Therefore, the number of clips sold in May is \( \frac{48}{2} = 24 \).

2. **Calculate the total number of clips sold in both months:**
   - Add the number of clips sold in April to the number of clips sold in May.
   - Total clips sold = Clips sold in April + Clips sold in May
   - Total clips sold = 48 + 24
   - Total clips sold = 72

Thus, the total number of clips Natalia sold in April and May is **\(\boxed{72}\).**

Figure 8: Case study on LLaDA-8B-Instruct Model with chain-of-thought reasoning problem.

**Prompt:** from typing import List

```
def has_close_elements(numbers: List[float], threshold: float) -> bool:
    """ Check if in given list of numbers, are any two numbers closer to each other than
    given threshold.
    >>> has_close_elements([1.0, 2.0, 3.0], 0.5)
    False
    >>> has_close_elements([1.0, 2.8, 3.0, 4.0, 5.0, 2.0], 0.3)
    True
```

| LLaDA-8B-Instruct | # 256 Decoding Steps |
| --- | --- |

Here's the implementation of the `has_close_elements` function:

```python
from typing import List

def has_close_elements(numbers: List[float], threshold: float) -> bool:
    """ Check if in given list of numbers, are any two numbers closer to
each other than
    given threshold.
    >>> has_close_elements([1.0, 2.0, 3.0], 0.5)
    False
    >>> has_close_elements([1.0, 2.8, 3.0, 4.0, 5.0, 2.0], 0.3)
    True
    """
    for i in range(len(numbers)):
        for j in range(i + 1, len(numbers)):
            if abs(numbers[i] - numbers[j]) < threshold:
                return True
    return False
```

| dParallel (Ours) | # 15 Decoding Steps |
| --- | --- |

```python
from typing import List

def has_close_elements(numbers: List[float], threshold: float) -> bool:
    """ Check if in given list of numbers, are any two numbers closer to
each other than
    given threshold.
    >>> has_close_elements([1.0, 2.0, 3.0], 0.5)
    False
    >>> has_close_elements([1.0, 2.8, 3.0, 4.0, 5.0, 2.0], 0.3)
    True
    """
    for i in range(len(numbers)):
        for j in range(i + 1, len(numbers)):
            if abs(numbers[i] - numbers[j]) < threshold:
                return True
    return False
```

Figure 9: Case study on LLaDA-8B-Instruct Model with naive code generation task.

**Prompt:** Write a function to check if the given number is woodball or not..

| LLaDA-8B-Instruct | # 256 Decoding Steps |
| --- | --- |

To determine if a given number is a "woodball," we need to define what a "woodball" is. For the sake of this example, let's assume a "woodball" is a number that is a multiple of 3. Here's a simple function in Python to check if a given number is a "woodball":

```python
def is_woodball(number):
    """
    Check if the given number is a 'woodball' (i.e., a multiple of 3).

    Parameters:
    number (int): The number to check.

    Returns:
    bool: True if the number is a 'woodball', False otherwise.
    """
    return number % 3 == 0

# Example usage:
print(is_woodball(9))  # True
print(is_woodball(5))  # False
```

In this function, `is_woodball`, we use the modulus operator `%` to check if the number is divisible by 3. If the number is divisible by 3 with no remainder, it is a "woodball," and the function returns `True`. Otherwise, it returns `False`.

| dParallel (Ours) | # 49 Decoding Steps |
| --- | --- |

Sure, I can help you with that. A woodball is a number that is a multiple of both 3 and 5. Here's a Python function to check if a given number is a woodball:

```python
def is_woodball(number):
    # Check if the number is a multiple of both 3 and 5
    if number % 3 == 0 and number % 5 == 0:
        return True
    else:
        return False

# Example usage:
print(is_woodball(15))  # Output: True
print(is_woodball(30))  # Output: True
print(is_woodball(10))  # Output: False
print(is_woodball(17))  # Output: False
```

This function checks if the number is divisible by both 3 and 5 using the modulus operator `%`. If both conditions are met, it returns `True`, indicating that the number is a woodball. Otherwise, it returns `False`.

Figure 10: Case study on LLaDA-8B-Instruct Model with instruction-based code generation task.

