# OpenReview forum: "dParallel: Learnable Parallel Decoding for dLLMs"
_ICLR.cc/2026/Conference — ICLR 2026 Poster_

### Official Review · Reviewer_FJ9p · 2025-10-27

**Soundness:** 2
**Presentation:** 2
**Contribution:** 2
**Rating:** 4
**Confidence:** 4

**Summary:**

This paper aims to further enhance the parallel decoding capability of Diffusion Large Language Models (DLLMs) through training. The authors argue that the suboptimal performance of current DLLMs in parallel decoding stems from their inadequate confidence distribution, which fails to support effective parallel generation. To address this, they propose a training strategy that explicitly encourages high-confidence predictions across multiple positions. Extensive experiments demonstrate that the trained model achieves significantly improved reasoning efficiency compared to the original model.

**Strengths:**

1. The acceleration of parallel decoding in DLLMs is a promising and worthwhile direction for further exploration.
2. Studying parallel decoding from the perspective of confidence distribution offers a novel and interesting angle.
3. The presented experimental results demonstrate a clear and substantial acceleration effect.

**Weaknesses:**

1. I have significant concerns regarding the motivation of the proposed method. For example, in Figure 3, the authors’ approach is based on the assumption that the confidence distribution shown in the lower figure is superior to that in the upper one. However, since a given prefix can often be completed in multiple reasonable ways, it is natural and intuitive for the confidence to be higher near the prefix. The authors should provide further clarification and evidence to justify this assumption.

2. I also have concerns regarding the role of Equation (7). Directly optimizing Equation (5) would inherently contribute to optimizing Equation (7) as well. Therefore, the authors should provide a more convincing explanation of the specific role and necessity of Equation (7), for instance, from an optimization perspective or through an analysis of the gradient behavior.

3. Since both the training and test datasets consist of similar mathematical and coding problems, I am concerned about whether the proposed method can truly serve as a generalizable paradigm for accelerating inference through fine-tuning, or if the observed acceleration merely results from overfitting to specific training data.

**Questions:**

See Weaknesses

---

> ### Author Response · Authors · 2025-11-23
> **Response to Reviewer FJ9p (1/2)**
>
> Thanks for the thoughtful and valuable feedback. Please find our responses below, we hope they adequately address your concerns. If you have any further questions or need additional clarification, we’d be happy to discuss them with you. Thanks again for your time and efforts.
>
> > **W1: Further clarification and evidence to justify the motivation of proposed method**
>
> **A:** Thanks for the professional comments. We agree that this point requires clearer explanation.
>
> For dLLMs, we consistently observe that confidence is a strong indicator of prediction correctness: low-confidence masked-token predictions are almost always low-quality. Empirically, reducing the decoding confidence threshold on LLaDA-8B-Instruct from 0.9 to 0.5 causes GSM8K accuracy to drop from 75.7% to 58.8%, confirming that low-confidence predictions cannot be safely used during decoding.
>
> As you noted, it is natural that only the 1–2 tokens closest to the prefix receive high confidence (Figure 3, upper plot). However, under this distribution, a dLLM can only make 1–2 reliable decisions per step, while the remaining masked tokens remain low-confidence and unusable, fundamentally limiting parallel decoding.
>
> To achieve highly parallel decoding, the model must be able to make high-quality decisions for multiple tokens at each step. High-quality decisions imply high confidence. Therefore, the key requirement is for dLLM to produce high-confidence predictions for multiple masked tokens simultaneously at every decoding step.
>
> Motivated by this, we propose certainty forcing, which trains the model to increase the confidence of all correct predictions within a block while keeping the original dLLM decoding trajectory unchanged. This approach enables the LLaDA model to make high-quality predictions for an average of 8-10 tokens per step, successfully realizing parallel decoding without compromising accuracy.
>
>
> > **W2: I also have concerns regarding the role of Equation (7). Directly optimizing Equation (5) would inherently contribute to optimizing Equation (7) as well.**
>
> **A:** Thanks for pointing this out. We appreciate the opportunity to clarify this further.
>
> Equation (5) is the standard cross-entropy (CE) loss, while Equation (7) is the entropy-minimization (EM) loss. The key difference is that CE mainly focuses on correctness. For non-target classes, its gradients are very small. During self-distillation, most tokens are already predicted correctly, so CE provides almost no additional force to further sharpen the distribution. In contrast, EM depends only on the shape of the probability distribution. It gives much larger gradients on non-target classes, forcing their probabilities downward and making the distribution sharper.
>
> For non-target classes, CE has gradient:
>
> $$\frac{\partial L_{CE}}{\partial z_k} = p_k ,$$
>
> where $z_k$ is the logit. When the model predicts the correct token, all non-target $p_k$ become small. The gradient therefore decreases linearly with $p_k$, leaving little incentive to increase confidence.
>
> For non-target classes, EM has gradient:
>
> $$\frac{\partial L_{EM}}{\partial z_k} = p_k (-\log p_k - H).$$
>
> When $p_k$ is small, the entropy term $H$ is negligible relative to $\log p_k$, giving:
>
> $$\frac{\partial L_{EM}}{\partial z_k} \approx p_k (-\log p_k).$$
>
> Compared to CE, EM introduces the extra $(-\log p_k)$ factor. As $p_k$ becomes small, this term grows rapidly, making EM’s gradients much larger than CE’s. As a result, even when the prediction is already correct, EM continues to strongly suppress non-target probabilities and significantly sharpens the distribution.
>
> Our ablation results in the paper confirm this effect. Under the same decoding-entropy threshold, models trained with EM have many more tokens achieving the threshold, leading to substantially higher parallel-decoding speedups.

---

> ### Author Response · Authors · 2025-11-23
> **Response to Reviewer FJ9p (2/2)**
>
> > **W3: I am concerned about whether the proposed method can truly serve as a generalizable paradigm for accelerating inference through fine-tuning, or if the observed acceleration merely results from overfitting to specific training data.**
>
> **A:** Thanks for the valuable feedback. To address this concern, we present three additional experimental results demonstrating that our method does not overfit to specific training data.
>
> First, we applied dParallel training to LLaDA-8B-Instruct using only 100k math problems. Despite training only on math domain, the model still performs strongly on out-of-domain coding tasks. As shown in Table 1, our method achieves substantial speedup while preserving accuracy on four coding benchmarks. This indicates that the model generalizes well.
>
> #### *Table 1. Trained on math data and Evaluation on coding bench. Accuracy and speedup are reported.*
> | Method | HumanEVAL| HumanEVAL++ | MBPP| MBPP++ |
> | - | - | -| -| -|
> | Original LLaDA  | 38.4% (1.0x) | 30.5% (1.0x)  | 42.4% (1.0x) | 57.9% (1.0x) |
> | dParallel LLaDA | 40.2% (8.2x) | 36.0% (7.6x)  | 40.8% (10.5x)| 57.1% (11.8x)|
>
> Second, we evaluate the dParallel LLaDA model on four standard general-purpose benchmarks: MMLU, ARC-C, HellaSwag, and PIQA. As shown in Table 2, our model maintains accuracy comparable to the original model, demonstrating that dParallel training does not harm general knowledge or language ability.
>
> #### *Table 2. Evaluation on general-purpose language tasks.*
> | Method | MMLU| HellaSwag | ARC-C| PIQA |
> | - | - | -| -| -|
> | Original  LLaDA  | 63.9 | 71.1 | 56.1 | 71.4|
> | dParallel LLaDA  | 63.7 | 70.7 | 56.8 | 71.9|
>
> Third, we further test the model on two open-ended generation tasks, TruthfulQA and CNN/DailyMail, covering open-domain QA and writing. As shown in Table 3, even on tasks where dLLMs are not particularly specialized, our approach still delivers clear speedup while keeping output quality close to the base model. Because open-ended tasks contain more ambiguity, the achievable speedup is naturally lower than on deterministic tasks such as math and code.
>
> #### *Table 3. Evaluation on open-ended QA and writing tasks. (length=128, blocksize=16)*
> | Method | TruthfulQA | CNN/DailyMail |
> | - | - | - |
> | Original  LLaDA  | 42.1 (1.0x) | 19.5 (1.0x) |
> | dParallel LLaDA  | 41.0 (4.2x) | 20.8 (2.0x) |
>
> In summary, our approach shows strong generalization and does not overfit to specific training data. It consistently provides substantial acceleration across diverse tasks while preserving accuracy, and the training procedure does not degrade the model’s general knowledge or capabilities.

---

> > ### Comment · Reviewer_FJ9p · 2025-11-25
> >
> > Thank you for the detailed response. While the additional experiments are helpful, I retain two concerns regarding the mechanism and the completeness of the results.
> >
> > 1. On Equation (7)
> >
> > I am not convinced that Equation (7) provides a distinct optimization objective. Fundamentally, CE and Equation (7) (EM) share the substantially same optimization direction. Minimizing CE against a one-hot target inherently minimizes the entropy of the predicted distribution. The "vanishing gradient" issue you mentioned relates to optimization dynamics rather than a fundamental difference in objectives. Equation (7) appears to merely amplify gradients for non-target classes (acting as a "sharpener") rather than introducing a distinct mechanism.
> >
> > So I believe there remains a significant gap in explaining the true source of the method's advantages.
> >
> > 2. Missing speedup metrics in Table 2. Evaluation on general-purpose language tasks.
> >
> > I noticed that Table 2 reports only Accuracy, unlike Tables 1 and 3 which explicitly include Speedup. Since the core contribution of this work is parallel decoding acceleration, omitting efficiency metrics for general benchmarks is concerning.
> >
> > Could you please provide the Speedup data for the benchmarks in Table 2? And how were the entropy thresholds set for the experiments in Table 2 and Table 3? Were they tuned specifically for each dataset to balance the trade-off, or was a fixed threshold applied across all tasks?

---

> > > ### Author Response · Authors · 2025-11-26
> > > **Response to Reviewer FJ9p**
> > >
> > > > **Q1:Explaining the advantage of EM loss.**
> > >
> > > Thanks for the thoughtful comments. We appreciate the opportunity to further clarify this important question.
> > >
> > > As you noted, the theoretical optimum of CE loss is indeed a one-hot distribution. However, in practice, the optimization dynamics prevent the model from reaching such high-confidence predictions. As discussed earlier, CE primarily focuses on correctness. Once the model predicts the correct class with moderate confidence, the gradient on the probability space $p$ decays linearly, and the gradient on the logit space $z$ decays exponentially. This exponential decay causes the gradient from correctly predicted tokens to become very small.
> > >
> > > As a result, the overall optimization direction is dominated by the small fraction of incorrectly predicted tokens in each batch. We measured the proportion of incorrect tokens during training and found that even in the final epoch, about 1% of tokens remain mispredicted in every batch. These few difficult tokens dominate the gradient signal throughout training, while the majority of already-correct tokens are stuck at “medium confidence” because their gradients are too small to be further optimized.
> > >
> > > In contrast, EM loss is agnostic to correctness. Even when a token is already predicted correctly, EM still provides sufficient gradients to push the distribution further into the low-entropy region.
> > >
> > > This is why our method requires both CE and EM losses. CE ensures the correctness of model prediction, while EM continuously drives the distribution toward low entropy. Using CE alone yields only limited speedup, and using EM alone leads to significant accuracy degradation.
> > >
> > >
> > > > **Q2:Missing speedup metrics in Table 2.**
> > >
> > > Thanks for the valuable feedback. This is indeed a point that requires further clarification.
> > >
> > > MMLU, ARC-C, HellaSwag, and PIQA are four widely used log-likelihood–based benchmarks from `lm-evaluation-harness`. Their prompts follow a multiple-choice format, and they are evaluated purely via log-likelihood scoring over four candidate options (“A/B/C/D”), there is no decoding or generation process involved. As a result, reporting speedup is not applicable for these benchmarks. To show the speedup across diverse tasks, we need to evaluate on generation-based benchmarks.
> > >
> > > As shown in Table 3, we have already demonstrated the speedup on two open-ended generation benchmarks, TruthfulQA and CNN/DailyMail. To further address your concerns, Table 4 reports additional experimental results on GPQA-CoT [1] and QASPER [2]. GPQA-CoT evaluates multi-disciplinary knowledge across chemistry, biology, and physics, while QASPER measures long-form question answering over academic papers. As shown in Table 4, our approach achieves significant speedup on both datasets while maintaining the accuracy, highlighting the generalization of our method across diverse tasks.
> > >
> > > For all experiments in Tables 1–4, we use a fixed entropy threshold = 0.5. Differences in speedup primarily stem from the inherent characteristics of each task. Highly formalized and deterministic tasks allow dLLMs to realize more parallel decoding potential, leading to larger gains. This observation is consistent with the current landscape of SOTA commercial dLLMs (e.g., Gemini-Diffusion, Mercury-Coder, Seed-Diffusion), which focus on code and math tasks where the efficiency advantage is most pronounced.
> > >
> > > #### *Table 4. Evaluation on GPQA-CoT and QASPER. (length=256, blocksize=32)*
> > > | Method | GPQA-CoT | QASPER |
> > > | - | - | - |
> > > | Original  LLaDA  | 25.2 (1.0x) | 11.6 (1.0x) |
> > > | dParallel LLaDA  | 25.9 (3.4x) | 11.7 (4.7x) |
> > >
> > > [1] "Gpqa: A graduate-level google-proof q&a benchmark." First Conference on Language Modeling. 2024.
> > >
> > > [2] "A dataset of information-seeking questions and answers anchored in research papers." arXiv:2105.03011 (2021).

---

> > > > ### Comment · Reviewer_FJ9p · 2025-11-28
> > > >
> > > > Thank you for the authors’ efforts. My view is that the authors’ explanation suggests the core issue is not that cross-entropy itself is a flawed optimization objective, but rather that CE is not fully or optimally optimized during the training dynamics. The introduction of the EM term appears to address this optimization issue of CE, rather than indicate that CE as a minimization objective is inherently problematic.
> > > >
> > > > Therefore, my suggestion is that the paper should reconsider how it motivates and introduces the method, so that readers can better grasp the true nature of the approach.
> > > >
> > > > Overall, I acknowledge the authors’ efforts, and I will increase my score to 6.

---

> > > > > ### Author Response · Authors · 2025-12-01
> > > > >
> > > > > Thanks for agreeing increasing the score and for your thoughtful comments. Your professional feedback has been truly valuable to us. We will carefully incorporate your suggestions to further improve our paper in the next version.

---

### Official Review · Reviewer_GE2Q · 2025-11-01

**Soundness:** 3
**Presentation:** 3
**Contribution:** 3
**Rating:** 6
**Confidence:** 4

**Summary:**

The paper introduces dParallel, a method to unlock the parallel decoding potential of diffusion large language models (dLLMs).
While dLLMs conceptually support parallel token prediction, existing models still require nearly token-length decoding steps to maintain quality.

To overcome the bottleneck that token confidence builds up left-to-right, they propose certainty-forcing distillation (CFD), a self-distillation technique that encourages the model to reach high certainty across masked tokens in parallel, without disrupting the original decoding trajectory.

When applied to LLaDA-8B-Instruct and Dream-7B-Instruct, dParallel dramatically reduces decoding steps (e.g., from 256 → 30 on GSM8K, ≈ 8.5× speedup) while maintaining or slightly improving accuracy.

**Strengths:**

1. The paper well identifies sequential certainty convergence as the fundamental bottleneck in parallel decoding is a sharp and well-motivated observation.

2. Certainty-forcing distillation is conceptually clear and easy to implement. It reuses a pre-trained dLLM as both teacher and student, requiring only modest computational cost (LoRA fine-tuning on consumer GPUs).

3. Across math and code benchmarks (GSM8K, MATH, MBPP, HumanEval), dParallel achieves 5–10× decoding acceleration with minimal or no accuracy drop.

4. The algorithm and implementation details (LoRA config, pseudocode, appendix) are precise and easy to follow and visualization of token confidence propagation (Figures 2 & 5) nicely supports the paper’s core claim about parallel convergence.

**Weaknesses:**

1. The “certainty-forcing” idea extends existing self-distillation and entropy-minimization concepts rather than introducing a wholly new framework. Its originality lies mainly in the application to dLLMs.

2. The method doesn’t fundamentally enhance reasoning or knowledge capability, and experiments rely on ~ 100 k self-generated math/code prompts; scaling behavior on larger, more diverse datasets or multilingual tasks remains unexplored.

3. The experiments  lack of natural-language, long-form, or open-ended generation tasks makes it unclear how general the method is beyond structured tasks.

4. While empirically convincing, there’s no formal discussion of how certainty dynamics relate to convergence or stability of diffusion decoding, which could also benefit to reveal why easoning or knowledge capability is not enhanced.

5. Recent works like Block Diffusion (Arriola et al. 2025) or Dimple (Yu et al. 2025b), which also target parallel decoding, are cited but not empirically compared.

**Questions:**

1. How does certainty-forcing behave on non-mathematical text (e.g., story generation or instruction following)?

2. Does parallel certainty hold for open-ended or ambiguous tokens?

3. The paper shows that confidence correlates with correctness, but could the model become over-confident and harm diversity?

4. How robust are results to the choice of β (balance weight) or temperature T in the entropy loss?

5. Is there a theoretical or empirical limit to how few decoding steps dLLMs can reach before degradation (e.g., 5–10 steps)?

---

> ### Author Response · Authors · 2025-11-23
> **Response to Reviewer GE2Q (1/3)**
>
> Thanks for the insightful and valuable feedback. We have provided detailed responses below and hope they address your concerns. We really appreciate your time and consideration.
>
> > **W1:The “certainty-forcing” idea extends existing self-distillation and entropy-minimization concepts rather than introducing a wholly new framework.**
>
> **A:** Thanks for the valuable comments. Our primary contribution identifies "sequential certainty convergence" as the critical bottleneck hindering the parallel decoding potential of dLLMs and argues that achieving "parallel certainty convergence" is essential. Accordingly, we innovatively introduce semi-ar forward masking and certainty loss in self-distillation. This straightforward approach enables dLLMs to generate 8–10 high-quality tokens per step, effectively translating theoretical parallel capabilities into reality. We believe this method helps close the efficiency gap between open-source and closed-source dLLMs, offering a fresh perspective and a strong baseline for future dLLM research.
>
> > **W2:The method doesn’t fundamentally enhance reasoning or knowledge capability.**
>
> **A:** Thanks for the valuable suggestions. dParallel is a post-training method designed specifically to unlock the parallel decoding potential of dLLMs and achieve significant inference acceleration. In the paper, we used 100k self-generated trajectories as training data to demonstrate that dParallel can improve parallel decoding and preserve accuracy without introducing any external knowledge.
>
> Following the reviewer’s advice, we further explored training with external data to enhance reasoning capability. We applied dParallel to LLaDA using math data generated by Dream, and found that this not only yielded a 6.4× speedup on GSM8K, but also boosted accuracy from 75.7% to 80.1% (Table 1). This highlights the strong potential of dParallel when trained on higher-quality external data. In future work, we plan to scale this approach to million-scale, more diverse datasets to further enhance the effectiveness of dParallel.
>
> #### *Table 1. Certrainty forcing training on LLaDA-8 with data generated by Dream-7B*
> | Method | GSM8K| MATH |
> | - | - | -|
> | Original Model | 75.7% (1.0×) | 33.5% (1.0×) |
> | dParallel Model| 80.1% (6.4×) | 31.9% (4.4×) |
>
> > **W3:The experiments lack of natural-language, long-form, or open-ended generation tasks.**
>
> **A:** Thanks for the professional comments. We evaluated the dParallel-trained LLaDA model on four general-purpose natural-language benchmarks (perplexity-based): MMLU, ARC-C, HellaSwag, and PIQA. As shown in Table 2, the model achieves performance comparable to the original, indicating that dParallel preserves the model’s general knowledge and language abilities
>
> #### *Table 2. Evaluation on general-purpose language tasks.*
> | Method | MMLU| HellaSwag | ARC-C| PIQA |
> | - | - | -| -| -|
> | Original LLaDA  | 63.9 | 71.1 | 56.1 | 71.4|
> | dParallel LLaDA  | 63.7 | 70.7 | 56.8 | 71.9|
>
> we further test the model on two open-ended generation tasks, TruthfulQA and CNN/DailyMail, covering open-domain QA and long-form summarization. As shown in Table 3, even on tasks where dLLMs are not particularly specialized, our approach still delivers clear speedup while keeping output quality close to the base model. Since open-ended tasks are inherently more ambiguous, the achievable speedup is naturally lower than on more deterministic tasks like math and code.
>
> #### *Table 3. Evaluation on open-ended generation tasks. (length=128, blocksize=16)*
> | Method | TruthfulQA | CNN/DailyMail |
> | - | - | - |
> | Original  LLaDA  | 42.1 (1.0x) | 19.5 (1.0x) |
> | dParallel LLaDA  | 41.0 (4.2x) | 20.8 (2.0x) |
>
> > **W4: Discussion of how certainty dynamics relate to convergence or stability of diffusion decoding.**
>
> **A:** Thanks for the valuable comments. During the decoding process, the dLLM's confidence for masked tokens initially oscillates before converging to a stable high value. This oscillation reflects a state of indecision, whereas convergence indicates that the model has finalized its decision. Notably, once the confidence converges, the predicted probability remains virtually invariant, meaning the model rarely reverses its decision.
>
> Our certainty-forcing method accelerates this convergence process without altering the final converged result. This is why the method significantly improves parallel decoding (by enabling the model to reach stable decisions earlier), yet it preserves the original accuracy instead of increasing it.

---

> ### Author Response · Authors · 2025-11-23
> **Response to Reviewer GE2Q (2/3)**
>
> > **W5: Recent works like Block Diffusion or Dimple, which also target parallel decoding, are cited but not empirically compared.**
>
> **A:** Thanks for the thoughtful comments. In Table 4, we compare our method with the parallel decoding approach proposed in Dimple. Our method achieves a significantly better speed–accuracy trade-off, reducing the number of decoding steps by a large margin while still maintaining accuracy.
>
> #### *Table 4. Comparing with the parallel decoding approach proposed in Dimple.*
> | Method | GSM8K| MATH | HumanEVAL| MBPP |
> | - | - | -| -| -|
> | Original LLaDA  | 75.7% (1.0×) | 33.5% (1.0×) | 38.4% (1.0×) | 42.4% (1.0×)  |
> | Dimple Decoding  | 75.5% (3.6×) | 33.2% (2.9×) | 37,2% (3.5×) | 41.6% (3.9×)  |
> | dParallel (ours) | 76.1% (8.5×) | 31.5% (5.7×) | 40.2% (8.2×) | 40.8% (10.5×) |
>
> We would also like to clarify the relationship between our work and block diffusion. Block diffusion is a new language-modeling paradigm positioned between dLLMs and AR-LLMs: it uses AR generation across blocks while applying diffusion within each block, enabling flexible sequence lengths, KV-cache usage, and the parallel decoding potential of dLLMs. Block diffusion is a different modeling paradigm that requires large-scale pretraining, while our method is a post-training technique designed to accelerate diffusion decoding.
>
> Therefore, the two are not competing approaches. Instead, they are complementary. Our method also has the potential to enhance the within-block parallelization of pretrained block diffusion models, which we believe is a promising future application.
>
>
> > **Q1: How does certainty-forcing behave on non-mathematical text.**
>
> **A:** Thanks for the comments. As shown in above Table 2, dParallel LLaDA achieves performance comparable to the original model across four ppl-based general benchmarks (MMLU, ARC-C, HellaSwag, PIQA), confirming that certainty-forcing preserves general language capabilities.
>
> Furthermore, above Table 3 demonstrates that on open-ended generation tasks (TruthfulQA, CNN/DailyMail), our approach delivers a clear speedup with negligible quality loss. While the inherent ambiguity of open-ended generation results in lower acceleration compared to deterministic tasks (e.g., math/code), the efficiency gains remain significant.
>
> > **Q2: Does parallel certainty hold for open-ended or ambiguous tokens?.**
>
> **A:** We appreciate the insightful comment. Even in open-ended generation, a significant portion of tokens remains deterministic. Our method leverages this by achieving parallel certainty convergence to accelerate the decoding of these tokens. However, open-ended tasks also contain many ambiguous tokens with high entropy, which hinders parallel convergence to high confidence levels. Consequently, as shown in Table 3, while our method achieves a clear speedup in open-ended tasks, the acceleration ratio is naturally lower compared to formalized Code and Math tasks.
>
> This result aligns with the current landscape of SOTA commercial dLLMs (Gemini-Diffusion, Mercury-Coder, Seed-Diffusion). These models all focus on Code and Math domains precisely because the efficiency advantage of dLLMs over AR-LLMs is most pronounced in such deterministic and formalized settings.
>
>
> > **Q3: The paper shows that confidence correlates with correctness, but could the model become over-confident and harm diversity?**
>
> **A:** Thanks for the professional comments. We agree that this point requires clearer clarification.
>
> We evaluated the generation diversity of dParallel LLaDA versus the original LLaDA on the TruthfulQA open-ended task. Specifically, we sampled 16 responses per prompt (temperature = 1.0) and measured diversity using two widely used metrics: Distinct-1 (lexical uniqueness) and Self-BLEU ($N$-gram overlap). As shown in Table 5, our method has a negligible impact on diversity compared to the baseline. The results confirm that our model remains in the optimal sweet spot, maintaining factually consistent while ensuring syntactically diverse.
>
> #### *Table 5. Evaluate diversity on open-ended generation tasks.*
> | Method | Distinct-1 ↑| Self-BLEU ↓|
> | - | - | - |
> | Original  LLaDA  | 0.187 | 0.603 |
> | dParallel LLaDA  | 0.188 | 0.661 |

---

> ### Author Response · Authors · 2025-11-23
> **Response to Reviewer GE2Q (3/3)**
>
> > **Q4: Impact of β (balance weight) or temperature T in the entropy loss?**
>
> **A:** Thanks for the valuable feedback. We conduct experiments to analyze the effects of `β` and `t`. (Table 6 and Table 7). All experiments are conducted with LLaDA-8B-Instruct, 4 training epochs, 100k training data, and 8 A5000GPUs.
>
> * **β** controls the weight of certainty loss. Larger `β` leading to higher speedup. When `β` is too large, it causes severe imbalance between certainty loss and consistency loss, therefore harming accuracy.
> * **t** rescales the token probability distribution before computing certainty loss. Smaller `t` weakens the certainty penalty, while overly large `t` again causes imbalance with consistency loss and reduces accuracy.
>
> Overall, `β` and `t` should be set so that the scales of certainty loss and consistency loss remain comparable. This balance is crucial for achieving parallel decoding gains without sacrificing accuracy. We set `β=2` and `t=0.5` in our final training.
>
> #### *Table 6. Impact of different β.*
> | Config | GSM8K| HumanEVAL |
> | - | - | -|
> | Original   | 75.7% (1.0×) |  38.4% (1.0×) |
> | β=1 t=0.5  | 76.4% (7.2×) |  39.0% (6.2×) |
> | β=2 t=0.5  | 76.0% (8.0×) |  39.6% (7.4×) |
> | β=4 t=0.5  | 74.2% (8.6×) |  36.0% (7.3×) |
> | β=8 t=0.5  | 73.2% (9.5×) |  33.5% (7.9×) |
>
> #### *Table 7. Impact of different t.*
> | Config | GSM8K| HumanEVAL |
> | - | - | -|
> | Original   | 75.7% (1.0×) |  38.4% (1.0×) |
> | β=2 t=0.5  | 76.0% (8.0×) |  39.6% (7.4×) |
> | β=2 t=1.0  | 69.0% (9.8×) |  31.7% (9.6×) |
> | β=2 t=2.0  | 63.7% (11.0×) |  20.7% (10.3×) |
>
>
> > **Q5: Is there a theoretical or empirical limit to how few decoding steps dLLMs can reach before degradation?**
>
> **A:** Thanks for the insightful comments. While there is no theoretical lower bound for decoding steps in DLLMs, empirical observations suggest task-dependent limits.
>
> For mathematical reasoning, the effective limit appears to be around 20-25 steps (reduced from 256), which we believe can be robustly achieved by scaling training data to the million level. For code generation, we believe the limit can be even lower, approximately 10-20 steps. This is because code is highly formalized and exhibits inherent structural parallelism, allowing for more aggressive step reduction compared to reasoning tasks.

---

> > ### Comment · Reviewer_GE2Q · 2025-11-26
> >
> > Thank you for the responses. After reading through the new results, I will keep my score as positive.

---

> > > ### Author Response · Authors · 2025-11-26
> > >
> > > We sincerely appreciate your recognition of our work and your thoughtful comments. Your valuable feedback has been immensely helpful to us. Should you have any further questions or concerns, we would be more than pleased to continue the discussion at any time.

---

### Official Review · Reviewer_Shkg · 2025-11-01

**Soundness:** 3
**Presentation:** 3
**Contribution:** 2
**Rating:** 6
**Confidence:** 2

**Summary:**

The paper introduces dParallel to address sequential certainty convergence in Diffusion LLMs. dParallel minimizes the entropy of top tokens to boost confidence in the model’s predictions, and makes inference semi-autoregressive by breaking down parallel decoding into blocks wherein tokens within a block are predicted in parallel, and blocks themselves follow autoregressive decoding. dParallel significantly reduces the latency in comparison to several recent works without taking a hit on accuracy.

**Strengths:**

* The latency reductions obtained by dParallel are substantial.
* The paper provides clear ablations on various components of the training algorithm.

**Weaknesses:**

* dParallel uses self distillation to train the LoRA parameters which requires running a large number of forward passes through the base model making training computationally expensive.

**Questions:**

* How would this fit with speculative decoding with a large autoregressive model (sharing the same vocabulary) since dParallel seems to satisfy all the requirements for a drafter.
* What would the final diffusion model look like if it was trained from scratch with dParallel’s objective (without the distillation)?
* How does LoRA compare to full finetuning for dParallel?

---

> ### Author Response · Authors · 2025-11-23
> **Response to Reviewer Shkg**
>
> Thanks for the thoughtful and valuable feedback. Please find our responses below, we hope they adequately address your concerns. If you have any further questions or need additional clarification, we’d be happy to discuss them with you. Thanks again for your time and efforts.
>
> > **W1: dParallel uses self distillation to train the LoRA parameters which requires running a large number of forward passes through the base model making training computationally expensive.**
>
> **A:** Thanks for raising this point, it is indeed a point that benefits from further clarification. While our approach utilizes self-distillation, which requires the original model to generate data, the cost of this data generation process is actually very low. For the LLaDA 8B model, utilizing the dInfer framework, all necessary data (100k) can be generated in just one day using 8 x A5000 GPUs(24G). Furthermore, our training process using LoRA is highly efficient, completing in 12 hours using 8 x A5000 GPUs.
>
> In addition, we want to emphasize that our dParallel method, as a post-training technique, is not only limited to self-distillation. It remains effective even when training on target data instead of data generated by the original model itself. To validate this general applicability, we conducted an experiment where we used responses generated by the DREAM model as target data to train the LLaDA model using our certainty-forcing approach. As shown in Table 1, our method not only achieves significant speedups but also yields over 4% accuracy improvements on GSM8K.
>
> #### *Table 1. Certrainty forcing training on LLaDA-8 with data generated by Dream-7B*
> | Method | GSM8K| MATH |
> | - | - | -|
> | Original Model | 75.7% (1.0×) | 33.5% (1.0×) |
> | dParallel Model| 80.1% (6.4×) | 31.9% (4.4×) |
>
>
> > **Q1: How would this fit with speculative decoding with a large autoregressive model?**
>
> **A:** Thanks for the insightful comments. We agree that this is a very promising application scenario for our method. Using dParallel, a small dLLM can be trained to act as a highly parallelized drafter. This drafter can decode an entire block of tokens at each step and pass these candidate tokens to a larger AR-LLM for confidence-based parallel verification. The AR-LLM then checks the block and triggers a rollback starting from the first token that is not accepted.
>
> With the AR-LLM serving as a reliable verifier, the dLLM can safely decode in a more parallel manner. We believe this speculative-decoding framework is indeed a reasonable and promising acceleration method.
>
>
> > **Q2: What would the final diffusion model look like if it was trained from scratch with dParallel’s objective?**
>
> **A:** Thanks for the insightful comments. We would like to further analyze the effect of dParallel in pretraining a dLLM from scratch.
>
> In the early stage of pretraining, we expect dParallel’s objective to behave very similarly to the standard dLLM training objective. This is because dParallel applies the certainty-forcing loss only on correctly predicted tokens, while the model is still primarily focused on memorizing new knowledge during the initial phase of pretraining.
>
> In the later stage of pretraining, the certainty-forcing loss should gradually become more effective. It encourages the model to generate multiple high-certainty tokens in a single step, ultimately producing a dLLM that can decode several masked tokens per step with consistently high confidence. Nevertheless, large-scale experiments would still be needed to fully understand whether the certainty-forcing loss might interfere with the model’s ability to memorize large amounts of knowledge during pretraining.
>
>
> > **Q3: How does LoRA compare to full finetuning for dParallel?**
>
> **A:** Thanks for the valuable comments. Using our 100k–sample training data, we find that LoRA achieves better overall performance than full finetuning. On LLaDA-8B-Instruct, LoRA yields an 8.5× speedup on GSM8K while maintaining 76.1% accuracy. In contrast, full finetuning increases the speedup to 11.0×, but accuracy drops to 66.7%.
>
> The main reason is that with only 100k training examples, full finetuning of an 8B-parameter model leads to overfitting and overly confident predictions, resulting in degraded accuracy. LoRA, by constraining the number of trainable parameters, avoids these issues and provides stronger performance at lower cost.

---

> ### Comment · Reviewer_Shkg · 2025-11-24
>
> Thank you for addressing my comments. I'll keep my score.

---

> > ### Author Response · Authors · 2025-11-24
> >
> > Thanks for your thoughtful reviews. We sincerely appreciate the time and care you have devoted to reviewing our work. Your detailed comments are extremely valuable in helping us improve our paper.

---

### Official Review · Reviewer_Aj7x · 2025-11-04

**Soundness:** 3
**Presentation:** 3
**Contribution:** 3
**Rating:** 6
**Confidence:** 3

**Summary:**

This paper proposes dParallel, a learnable method to accelerate dLLMs by reducing the number of required decoding steps. The authors first identify a key bottleneck to parallel generation: the "sequential certainty convergence," where the model's confidence in its predictions propagates from left-to-right rather than emerging in parallel. To address this, they introduce "certainty-forcing distillation," a training strategy that fine-tunes a pre-trained dLLM using a new objective. This objective combines a standard consistency loss (to follow the original model's generation trajectories) with a certainty-forcing loss (an entropy minimization term) that encourages the model to become highly confident in its correct predictions more rapidly. The method is evaluated on top of two SOTA academia-setting models LLaDA and Dream models, shwoing significant reductions in decoding steps (e.g., 8.5x on GSM8K) while maintaining the original model's accuracy.

However, I am recommending a borderline score at this time due to two major reservations that temper the paper's claims and require clarification:
1.  **Conflation of Training and Inference Strategies:** The experimental setup makes it difficult to isolate the true benefit of the proposed *training* method. The dParallel-trained model is evaluated with a specific entropy-based sampler, while the baselines (which are untrained) use different samplers. The gains may be attributable to the superiority of the entropy sampler itself, rather than the certainty-forcing distillation. (I could be wrong but feel free to correct me on this authors)

2.  **Concerns About Specialized Fine-Tuning:** The distillation process uses a relatively small (100k samples) dataset generated from specific domains (e.g., math and code). This raises the concern that dParallel is effectively a form of specialized fine-tuning, and its impressive performance might not generalize to broader, out-of-domain tasks.

I am open to increasing my score if the authors can provide clean explanations or cleanly ablate the effects of their training method and demonstrate its generalizability beyond the distillation domains or have discussions on addressing this weakness too.

**Strengths:**

1. The identification and visualization of "sequential certainty convergence" is a clear and valuable contribution that helps formalize why parallel decoding in dLLMs is challenging.

2. The certainty-forcing loss, which minimizes entropy only on correctly predicted tokens, is an elegant way(IMO) to encourage higher confidence without penalizing the model during its exploration phase.

3. The reported speedups (8.5x-10.5x) with little to no performance drop on challenging reasoning benchmarks are substantial and demonstrate the high potential of the approach.

**Weaknesses:**

- **Experimental Design Conflates Variables:** This is the most critical weakness. The paper compares `(dParallel-trained model + entropy sampler)` against `(original model + other samplers)`. To truly validate the contribution of the *certainty-forcing distillation*, the essential baseline is missing: `(original model + entropy sampler)`. Without this comparison, it is impossible to disentangle the gains from the training method versus the gains from the inference-time sampler. It is plausible that the entropy sampler is simply a better fit for dLLMs, and the training method provides only a marginal extra benefit.

- **Overfitting danger**: The distillation data is sourced from math and code problems. While the main evaluation is on math and code benchmarks, this raises a serious question about generalization. Does this specialized training harm the model's performance on general-purpose language tasks like commonsense reasoning (e.g., MMLU, HellaSwag) or instruction following? The lack of evaluation on these standard benchmarks is a major omission and leaves the true "cost" of this training unclear.

-   **Insufficient Ablation of the Method:** The paper ablates the training objective components but provides limited analysis on key design choices. For example, the certainty loss is applied only to tokens the model already predicts correctly. What percentage of tokens does this typically affect during training? How does this dynamic change over the course of training? Furthermore, the impact of the temperature `T` and the loss weight `β` are not explored, making it hard to assess the method's sensitivity. I understand that one paper might not cover everything but it is worth including some sort discussions and inform the future researchers who might be using this method.

**Questions:**

1.  To isolate the effect of your proposed training method, could you provide results for the original, untrained LLaDA model using the *same entropy-threshold sampler* you use for dParallel? This is a crucial experiment to demonstrate that the gains are primarily from certainty-forcing distillation and not just the choice of sampler.

2.  Have you evaluated dParallel on broader, general-purpose benchmarks (e.g., MMLU, ARC, HellaSwag) to confirm that the specialized distillation on math/code data does not cause performance regression on out-of-domain tasks?

3.  Regarding the certainty-forcing loss (Eq. 7), what is the typical size of the set `Mc` (correctly predicted tokens) relative to `Ma` (all masked tokens) during training, which kinda of distribution does it follow w.r.t. `t` ? I might miss some details feel free to correct me? Does this ratio change significantly, and how does it affect the stability and effectiveness of the learning signal?

4.  The paper argues that consistency distillation alone is insufficient. However, in Table 1, the performance drop for "Consistency Distillation" is quite severe (e.g., 75.7% -> 69.9% on GSM8K). This seems worse than simple few-step decoding. Could you clarify why this baseline performs so poorly? Does this suggest a potential issue with the self-distillation setup itself?

Minor questions:
- some discussion of the semi-autoregressive masking strategy during training would be beneficial. Why was a block size of 32 chosen for LLaDA but 256 for Dream? How does this choice interact with the goal of promoting parallel certainty?

---

> ### Author Response · Authors · 2025-11-23
> **Response to Reviewer Aj7x (1/2)**
>
> Thanks for the insightful and valuable feedback. We have provided detailed responses below and hope they address your concerns. We really appreciate your time and consideration.
>
> > **W1 Q1: disentangle the gains from the training method versus the gains from the inference-time sampler.**
>
> **A:** Thanks for the valuable comments. We agree that further clarification is necessary. To address your concern, we provide Table 1 below, which compares the performance of the `original model`, the `original model + entropy sampler`, and `dParallel model + entropy sampler`.
>
> As shown in the Table 1, the model trained with our certainty-forcing method (dParallel) significantly outperforms `original model + entropy sampler` across all benchmarks. It achieves substantially higher speedups while maintaining accuracy. This result demonstrates that the performance gains primarily stem from our new training strategy rather than the sampling method itself.
>
> #### *Table 1. Disentangling the gains from the entropy sampler. We report accuracy and speedup. The entropy threshold is set as 0.5.*
> | Method | GSM8K| MATH | HumanEVAL| MBPP |
> | - | - | -| -| -|
> | Original LLaDA Model | 75.7% (1.0×) | 33.5% (1.0×) | 38.4% (1.0×) | 42.4% (1.0×)  |
> | Original LLaDA Model + entropy-sampler  | 75.4% (3.3×) | 33.4% (2.6×) | 32.9% (3.1×) | 40.8% (3.3×)  |
> | dParallel LLaDA Model + entropy-sampler | 76.1% (8.5×) | 31.5% (5.7×) | 40.2% (8.2×) | 40.8% (10.5×) |
>
>
> > **W2 Q2: Does this specialized training harm the model's performance on general-purpose language?**
>
> **A:** Thanks for the insightful comments. Following your suggestion, we evaluated our dParallel LLaDA model on general-purpose benchmarks: MMLU, HellaSwag, ARC-C and PIQA. As shown in Table 2, our method maintains comparable performance to the original model across these diverse language tasks. This indicates that our training strategy does not come at the cost of the model’s general language capabilities.
>
> #### *Table 2. Evaluation on general-purpose language tasks.*
> | Method | MMLU| HellaSwag | ARC-C| PIQA |
> | - | - | -| -| -|
> | Original LLaDA  | 63.9 | 71.1 | 56.1 | 71.4|
> | dParallel LLaDA  | 63.7 | 70.7 | 56.8 | 71.9|
>
>
> > **W3 Q3: Insufficient Ablation of the Method.**
>
> **A:** Thanks for the valuable suggestions. During training, the Mc/Ma ratio remains very high. At the start of training, Mc/Ma is 98.38% (first 128 iterations), and after one epoch it increases to 99.17%. Although this affects only about 1% of the training tokens, this small portion has a significant impact on the final outcome. Table 3 compares applying certainty loss to all tokens (Ma) versus only to correct predicted tokens (Mc). When applied to all tokens, accuracy drops and training becomes unstable. This confirms that certainty loss should be applied selectively.
>
> #### *Table 3. Certainty-Forcing loss on different tokens.*
> | Method | GSM8K| HumanEVAL |
> | - | - | -|
> | Original LLaDA  | 75.7% (1.0×) |  38.4% (1.0×) |
> | CF loss on all tokens  | 72.3% (8.9×) |  36.0% (7.9×) |
> | CF loss on correct predicted tokens  | 76.1% (8.5×) |  40.2% (8.2×) |
>
> We also conduct experiments to analyze the effects of `β` and `t`. (Table 4 and Table 5). All experiments are conducted on LLaDA-8B-Instruct with 4 training epochs, 100k training data, and 8 A5000GPUs.
>
> * **β** controls the weight of certainty loss. Larger `β` leading to higher speedup. When `β` is too large, it causes severe imbalance between certainty loss and consistency loss, therefore harming accuracy.
> * **t** rescales the token probability distribution before computing certainty loss. Smaller `t` weakens the certainty penalty, while overly large `t` again causes imbalance with consistency loss and reduces accuracy.
>
> Overall, `β` and `t` should be set so that the scales of certainty loss and consistency loss remain comparable. This balance is crucial for achieving parallel decoding gains without sacrificing accuracy. We set `β=2` and `t=0.5` in our final training.
>
> #### *Table 4. Impact of different β.*
> | Config | GSM8K| HumanEVAL |
> | - | - | -|
> | Original   | 75.7% (1.0×) |  38.4% (1.0×) |
> | β=1 t=0.5  | 76.4% (7.2×) |  39.0% (6.2×) |
> | β=2 t=0.5  | 76.0% (8.0×) |  39.6% (7.4×) |
> | β=4 t=0.5  | 74.2% (8.6×) |  36.0% (7.3×) |
> | β=8 t=0.5  | 73.2% (9.5×) |  33.5% (7.9×) |
>
> #### *Table 5. Impact of different t.*
> | Config | GSM8K| HumanEVAL |
> | - | - | -|
> | Original   | 75.7% (1.0×) |  38.4% (1.0×) |
> | β=2 t=0.5  | 76.0% (8.0×) |  39.6% (7.4×) |
> | β=2 t=1.0  | 69.0% (9.8×) |  31.7% (9.6×) |
> | β=2 t=2.0  | 63.7% (11.0×) |  20.7% (10.3×) |

---

> ### Author Response · Authors · 2025-11-23
> **Response to Reviewer Aj7x (2/2)**
>
> > **Q4: Why consistency distillation not works on dLLM.**
>
> **A:**  Thanks for the thoughtful comments. We would like to offer a further analysis of this phenomenon. As shown in Table 1, compared to the original model's direct few-step decoding, Consistency Distillation yields only a modest accuracy improvement on GSM8K (from 68.6% to 69.9%) at 64 decoding steps.
>
> The main reason is that the pretraining objective of dLLM is already similar to that of consistency models. dLLMs are pre-trained by randomly masking tokens in a clean sequence and training the model to reconstruct the fully unmasked tokens directly. This 'direct prediction' approach aligns closely with the training objective of consistency models.
>
> Consequently, the main distinction between dLLM pre-training and Consistency Distillation lies merely in the source of the masked input: pre-training uses random masking, whereas Consistency Distillation samples a realistic intermediate state from the decoding trajectory. Because the pre-trained dLLM is already pretrained on direct reconstruction, the performance gain provided by Consistency Distillation is naturally limited. That's why we need to provide a new training method for the few-step dLLM.
>
>
> > **Q5: Why was a block size of 32 chosen for LLaDA but 256 for Dream? .**
>
> **A:** Thanks for the comments. The main reason is that the Dream model is adapted from an AR-LLM. In the semi-AR forward process, using a small block size creates an input sequence that closely resembles the input format of the original AR-LLM. Training on these  AR-like masked sequences causes Dream to gradually regress back toward an AR-LLM, which in turn lowers its accuracy in the diffusion setting. In contrast, the LLaDA model, which is trained from scratch, does not suffer from this issue as it lacks the initial AR-LLM bias.
>
> Therefore, for DREAM, we are required to use a relatively large block size (256) during training to mitigate this degradation. For LLaDA, we have the flexibility to use a flexible block size, and during training, we specifically used 32, which is the same block size employed during its semi-AR inference.

---

> ### Author Response · Authors · 2025-11-24
> **Additional response to Reviewer Aj7x**
>
> > **Generalizability beyond the distillation domains**
>
> Thanks for the professional comments. To address the concerns, we present additional experimental results to demonstrate our method's generalizability beyond the distillation domains.
>
> First, we applied dParallel training to LLaDA-8B-Instruct using only 100k math problems. Despite training only on math domain, the model still performs strongly on out-of-domain coding tasks. As shown in Table 6, our method achieves substantial speedup while preserving accuracy on four coding benchmarks. This indicates that the model generalizes well and does not overfit to specific training data.
>
> #### *Table 6. Trained only on math data and Evaluation on coding bench. Accuracy and speedup are reported.*
> | Method | HumanEVAL| HumanEVAL++ | MBPP| MBPP++ |
> | - | - | -| -| -|
> | Original LLaDA  | 38.4% (1.0x) | 30.5% (1.0x)  | 42.4% (1.0x) | 57.9% (1.0x) |
> | dParallel LLaDA | 40.2% (8.2x) | 36.0% (7.6x)  | 40.8% (10.5x)| 57.1% (11.8x)|
>
> Second, we further test the model on two open-ended generation tasks, TruthfulQA [1] and CNN/DailyMail [2], covering open-domain QA and writing. As shown in Table 7, even on tasks where dLLMs are not particularly specialized, our approach still delivers clear speedup while keeping output quality close to the base model. Because open-ended tasks contain more ambiguity, the achievable speedup is naturally lower than on deterministic tasks such as math and code.
>
> #### *Table 7. Evaluation on open-ended QA and writing tasks. (length=128, blocksize=16)*
> | Method | TruthfulQA | CNN/DailyMail |
> | - | - | - |
> | Original  LLaDA  | 42.1 (1.0x) | 19.5 (1.0x) |
> | dParallel LLaDA  | 41.0 (4.2x) | 20.8 (2.0x) |
>
> Table 8 reports additional experimental results on GPQA-CoT [3] and QASPER [4]. GPQA-CoT evaluates multi-disciplinary knowledge across chemistry, biology, and physics, while QASPER measures long-form question answering over academic papers. As shown in Table 8, our approach achieves significant speedup on both datasets while maintaining the accuracy.
>
> #### *Table 8. Evaluation on GPQA-CoT and QASPER. (length=256, blocksize=32)*
> | Method | GPQA-CoT | QASPER |
> | - | - | - |
> | Original  LLaDA  | 25.2 (1.0x) | 11.6 (1.0x) |
> | dParallel LLaDA  | 25.9 (3.4x) | 11.7 (4.7x) |
>
> For all experiments in above Tables, we use a fixed entropy threshold = 0.5. Differences in speedup primarily stem from the inherent characteristics of each task.
>
> In summary, our approach shows strong generalization and does not overfit to specific training data. It consistently provides substantial acceleration across diverse tasks while preserving accuracy, and the training procedure does not degrade the model’s general knowledge or capabilities.
>
>
> [1] "Truthfulqa: Measuring how models mimic human falsehoods." Proceedings of the 60th annual meeting of the association for computational linguistics. 2022.
>
> [2] "Teaching machines to read and comprehend." Advances in neural information processing systems 28 (2015).
>
> [3] "Gpqa: A graduate-level google-proof q&a benchmark." First Conference on Language Modeling. 2024.
>
> [4] "A dataset of information-seeking questions and answers anchored in research papers." arXiv:2105.03011 (2021).

---

### Author Response · Authors · 2025-12-01
**Review Discussion Summary for Submission 2542**

Dear Chairs,

We would like to express our sincere appreciation for the time, effort, and thoughtful consideration that you and all reviewers have devoted to evaluating our submission. We are truly grateful for the constructive feedback and for the reviewers’ recognition of the novelty of our method, the clarity of our contributions, the strong experimental effectiveness, the simplicity of our implementation, and the potential impact of our work.

Reviewers **Aj7x, Shkg, and GE2Q** initially provided positive scores. During the discussion phase, reviewers **Shkg** and **GE2Q** indicated that our rebuttal addressed their concerns and confirmed that they would keep the positive score. Reviewer **Aj7x** also wrote that he was “open to increasing my score”, but he was unable to join the discussion before it closed.

Reviewer **FJ9p** initially gave a score of 4, with two main questions:
(1) why entropy minimization (EM) can effectively encourage high-confidence predictions for dLLMs while cross-entropy (CE) may fail to do so, and
(2) whether the proposed dParallel training might overfit to specific data.

- To address (1), we conducted a comprehensive gradient analysis of EM loss and CE loss, providing a theoretical explanation for the optimization dynamics, which is also strongly supported by the ablation studies in our paper.

- To address (2), we performed additional experiments showing that a dLLM trained with dParallel solely on mathematical data can still achieve significant speedup (average 6.6x) while maintaining original accuracy across diverse out-of-domain generation tasks. These results demonstrate our strong generalization.

The additional analyses and experiments resolved reviewer **FJ9p**’s concerns, and he subsequently raised his score to a positive 6.

We are deeply thankful for the constructive engagement throughout the review process, and we hope that the above information is helpful in your final assessment.

With sincere gratitude,

Authors of submission 2542

---

### Meta-Review · Area_Chair_UPBr · 2026-01-08

**Summary:**

This paper proposes dParallel that enables parallelism of diffusion LLMs for efficient decoding. The core technical contribution is certainty-forcing distillation that distills the model to follow its original sampling trajectories while achieving high certainty on masked tokens so that the number of decoding steps can be reduced and performance is preserved. The authors demonstrated consistent speedups of dParallel over baselines across a set of commonly used LLM benchmarks.

The reviewers were generally positive about this work. While there were initial concerns regarding some technical details and evaluation on general language tasks to validate generalization, the authors did a good job addressing these concerns in the rebuttal, and all reviewers seemed positive about this work post-rebuttal.

**Reviewer Concerns:**

Reviewer FJ9p indicated that their concerns were addressed and were willing to raise the score. All the other reviewers' concerns were relatively minor and addressed well.

**Reviewer Scores:**

The only slightly negative reviewer explicitly indicated that they were willing to increase the score, and thus all reviewers would have been positive about this submission.

---

### Decision · Program_Chairs · 2026-01-26

Accept (Poster)